# Introducing hydrometeor orientation into all-sky microwave/sub-millimeter assimilation

Vasileios Barlakas[1], Alan J. Geer[2], and Patrick Eriksson[1]

[1]Department of Space, Earth and Environment, Chalmers University of Technology, Gothenburg, Sweden
[2]European Center for Medium-Range Weather Forecasts, Reading, UK

**Correspondence:** Vasileios Barlakas (vasileios.baralakas@chalmers.se)

**Abstract.** Numerical weather prediction systems still employ many simplifications when assimilating microwave radiances in all-sky conditions (clear sky, cloudy, and precipitation). For example, the orientation of ice hydrometeors is ignored, along with the polarization that this causes. We present a simple approach for approximating hydrometeor orientation, requiring minor adaption of software and no additional calculation burden. The approach is introduced in the RTTOV (Radiative Transfer for TOVS) forward operator and tested in the Integrated Forecast System (IFS) of the European Centre for Medium-Range Weather Forecasts (ECMWF). For the first time within a data assimilation (DA) context, this represents the ice induced brightness temperature differences between vertical (V) and horizontal (H) polarization, the polarization difference (PD). The discrepancies in PD between observations and simulations decrease by an order of magnitude at 166.5 GHz, with maximum reductions of 10–15 K. The error distributions, which were previously highly skewed and therefore problematic for DA, are now roughly symmetrical. The approach is based on rescaling the extinction in V- and H-channels, which is quantified by the polarization ratio $\rho$. Using dual polarization observations from Global Precipitation Mission microwave imager (GMI), suitable value for $\rho$ was found to be 1.5 and 1.4 at 89.0 and 166.5 GHz, respectively. The scheme was used for all the conical scanners assimilated at ECMWF, with broadly neutral impact on the forecast, but with an increased physical consistency between instruments that employ different polarizations. This opens the way towards representing hydrometeor orientation for cross-track sounders, and at frequencies above 183.0 GHz where the polarization can be even stronger.

*Copyright statement.* TEXT

## 1 Introduction

Clouds containing ice hydrometeors are considered among the greatest ambiguities in both climate and numerical weather prediction (NWP) modeling systems (Duncan and Eriksson, 2018). Despite recent significant progress, their interaction with radiation has not been reliably assessed, in part owing to their high heterogeneity in shape and orientation within a cloud.

Over the last decades, several studies explored the use of polarimetric observations at millimeter/sub-millimeter (mm/sub-mm) wavelengths to retrieve microphysical and macrophysical properties of ice hydrometeors (e.g., Czekala, 1998; Prigent

et al., 2005; Xie, 2012; Defer et al., 2014). In particular, and the focus of this study, oriented non-spherical ice hydrometeors are known to cause the observed brightness temperature ($T_{\mathrm{B}}$) differences between vertical (V) and horizontal (H) polarization.

Here, this is denoted as the polarization difference (PD = $T_{\mathrm{BV}} - T_{\mathrm{BH}}$), owing to different scattering properties between V- and H-polarization (dichroism effect, Davis et al., 2005). Evans and Stephens (1995a, b) simulated the sensitivity of polarized microwave (MW) frequencies (between 85.5 and 340.0 GHz) on non-spherical horizontally oriented ice hydrometeors. They reported a positive polarization signal, which increases with frequency. These findings were further supported by Czekala (1998); by conducting off-nadir simulations including polarization at 200.0 GHz, he reported a PD of about 30 K linked to

irregular shaped horizontally oriented hydrometeors. Xie (2012) reported a PD up to 10 K due to oriented snow observed by a ground-based passive radiometry at 150.0 GHz. Studies involving active and passive satellite remote sensing instruments further supported the presence of oriented ice hydrometeors (e.g., Prigent et al., 2001, 2005; Davis et al., 2005; Defer et al., 2014; Gong and Wu, 2017; Gong et al., 2020). Prigent et al. (2001, 2005) focused on passive MW imagers operating at low frequencies (37.0 and 85.0 GHz), i.e., the Tropical Rainfall Measuring Mission's (TRMM) Microwave Imager (TMI)

and the Special Sensor Microwave/Imager (SSM/I), and reported a PD due to horizontally oriented liquid at 37.0 GHz or ice hydrometeors at 85.0 GHz. At higher frequencies (up to 166.5 GHz), analogous PDs have been reported by Defer et al. (2014) and Gong and Wu (2017) derived from the Microwave Analysis and Detection of Rain and Atmospheric Structure (MADRAS) and the GPM (Global Precipitation Mission) microwave imager (GMI), respectively.

  Defer et al. (2014) and Gong and Wu (2017) tried to interpret the observed PD in a conceptual way. Especially, Gong and

Wu (2017) suggested an approximative modeling framework linking the observed PD to the ratio of optical thicknesses in V- and H-polarization ($\tau_{\mathrm{H}}/\tau_{\mathrm{V}}$); this ratio is linked to the axial ratio of an oriented hydrometeor. This framework was tested on GMI observations at 89.0 and 166.5 GHz, but also on observations by the airborne radiometer CoSSIR (Compact Scanning Submillimeter-wave Imaging Radiometer) at a frequency of 640.0 GHz. The first rigorous simulations reproducing the observed PD were conducted by Brath et al. (2020). One of their main findings was the strong impact of the assumed shape

of the ice hydrometeor on the simulated PD. Presently, GMI is the only operational instrument that measures ice hydrometeors including polarization. The upcoming Ice Cloud Imager (ICI) mission (Eriksson et al., 2020) will cover mm/sub-mm frequencies and will have channels measuring both V- and H- polarization at 243.2 and 664.4 GHz. This will provide further insights regarding the strong polarization signals originated by oriented ice hydrometeors. The observations from ICI will be assimilated in all-sky conditions for weather forecasting, further motivating the need to handle hydrometeor orientation.

All-sky radiance assimilation is the process of assimilating observations under the complete range of atmospheric conditions, i.e., clear sky, cloudy, and precipitation. Microwave humidity satellite observations in particular are gaining weight in forecasts, with the possibility to infer improved dynamical initial conditions (e.g., temperature, winds) from observations of cloud and precipitation (Geer et al., 2018). At the European Centre for Medium-Range Weather Forecasts (ECMWF), all-sky observations comprise around 20 % of the observation impact on forecasts (Geer et al., 2017). There is an ongoing effort to incorporate all

possible observations that are sensitive to cloud and precipitation through the development of new capabilities, i.e., including visible, infrared, and sub-mm wavelengths (Geer et al., 2018).

The assimilation of satellite observations necessitates accurate, and at the same time, fast radiative transfer simulations. Currently, the leading radiative transfer models that are employed in global NWP systems for data assimilation (DA) are the Radiative Transfer model for TIROS Operational Vertical Sounder (RTTOV, Saunders et al., 2018) and the Community Radiative Transfer Model (CRTM, Liu and Boukabara, 2014). However, polarization due to the orientation of non-spherical hydrometeors is currently ignored. These forward models apply only "scalar" simulations; one calculation deals with either V- or H-polarization, while in nature scattering by hydrometeors transfers energy between polarizations. In addition, the models use the same ice scattering properties for both computations. It is currently too expensive to move to fully-polarized radiative transfer in an assimilation system. However, this work will show that even with scalar radiative transfer, a significantly improved physical representation can be achieved by correctly varying the scattering properties as a function of polarization.

Following the work of Gong and Wu (2017), we explore the use of a simple scheme to approximate the effect of oriented ice hydrometeors in reproducing the observed PD from conical scanning MW imagers. In this study, a special emphasis is given on quantifying the best fit between model and observations. The performance of such scheme is also tested in cycled DA experiments by utilizing the Integrated Forecast System (IFS) of ECMWF. An ability to model oriented hydrometeors will provide essential information towards assimilating the observations from upcoming satellite missions, e.g., ICI and the Microwave Imager (MWI) that will fly on board the Meteorological Operational Satellite-Second Generation (Metop-SG) operated by the European Organization of Meteorological Satellites (EUMETSAT). Reducing errors in forward modelling could improve the quality of the initial conditions for weather forecasting, as well as further improve weather forecasts.

## 2   Methods and tools

### 2.1   Radiative transfer model

Radiative transfer simulations are conducted by means of RTTOV-SCATT (Bauer et al., 2006) that accounts for multiple scattering radiative transfer at MW and sub-mm frequencies and is part of the RTTOV package (Saunders et al., 2018). RTTOV-SCATT was developed by the EUMETSAT Numerical Weather Prediction Satellite Application Facility (NWP SAF) and is utilized by weather centres and scientists worldwide for fast modelling of all-sky radiances, including as the observations operator in the IFS at ECMWF.

RTTOV-SCATT employs the delta-Eddington approximation (Joseph et al., 1976) to solve the radiative transfer equation in all-sky conditions (clear, cloudy, and precipitating). The simulated $T_\mathrm{B}$ is calculated as the weighted mean of the $T_\mathrm{B}$ from two independent columns linked to clear-sky ($T_\mathrm{B,clear}$) and cloudy ($T_\mathrm{B,cloud}$) conditions:

$$T_\mathrm{B} = (1 - C_\mathrm{w}) \cdot T_\mathrm{B,clear} + C_\mathrm{w} \cdot T_\mathrm{B,cloud}. \tag{1}$$

Where $C_\mathrm{w}$ is the effective cloud fraction representing the vertical mean of cloud and precipitation fractions weighted by the hydrometeor content (Geer et al., 2009a, b). This is considered to be a fast approximation of sub-grid variability and hence, the beam-filling effect (e.g., Barlakas and Eriksson, 2020).

The gaseous absorption component (e.g., oxygen and water vapour) is derived from precalculated regression tables supplied by the core RTTOV model (Saunders et al., 2018). Here, we employ version 12.3 (v12.3) of RTTOV-SCATT, but with two main modifications on the path towards the final version 13 (v13.0) configuration. First, instead of a four hydrometeor configuration (rain, snow, cloud liquid water, and cloud ice water), it separates convective snow (ice hydrometeors in deep convective cores; hereafter graupel) from large scale snow (precipitating ice hydrometeors in stratiform clouds; hereafter snow) leading to a five hydrometeor convention; a feature not available in v12.3. Further clarifications are given in Sect. 2.2. One of the main new aspects is the availability of a wider range of particle size distributions (PSD) (e.g., McFarquhar and Heymsfield, 1997; Petty and Huang, 2011; Heymsfield et al., 2013) and the Atmospheric Radiative Transfer Simulator (ARTS) single scattering database (Eriksson et al., 2018). This database is comprised of more realistic hydrometeor shapes (e.g., hail, graupel, and aggregates) covering a greater frequency range (1.0 to 886.4 GHz) compared to Liu (2008) and hence, ideal for non-spherical hydrometeors; especially, for the upcoming ICI mission. RTTOV-SCATT utilizes pre-calculated bulk optical properties (extinction, single scattering albedo, and asymmetry parameter) for each hydrometeor (rain, snow, graupel, cloud liquid water, and cloud ice water), i.e., the hydrometeor tables. For each model level, the bulk properties are derived from the hydrometeor tables given the hydrometeor content and temperature. For details the reader is referred to Geer and Baordo (2014).

Over ocean, surface emissivity is calculated by the FAST microwave Emissivity Model-6 (FASTEM-6, Kazumori and English, 2015). The land surface emissivity is primarily retrieved from satellite observations in surface sensitive channels following the emissivity retrieval approach described in Karbou et al. (2010). This approach is based on the assumption that surface emissivity varies only slowly with frequency. Accordingly, surface emissivities are first retrieved in window channels at lower frequencies and then these values are used as the surface emissivity for nearby sounding channels. This approach was extended to all-sky assimilation by Baordo and Geer (2016). However, the retrieval can be unreliable in strongly scattering scenes, if the cloud and precipitation is inconsistent between model and observations; it can generate a surface emissivity outside the physical bounds of 0 and 1, and even if within those bounds, the retrieval must be further quality checked against values from an atlas. The atlas values are from the Tool to Estimate Land Surface Emissivities at Microwave (TELSEM) (Aires et al., 2011), which is a monthly average emissivity climatology constructed from 10 years observations from the Special Sensor Microwave/Imager (SSM/I). If the emissivity retrieval is out of physical bounds or far from the value in the atlas, the atlas value is used instead.

## 2.2 Microphysical configuration

The microphysical setup employed in this study is found in Table 1. This setup was derived by Geer (2021, referred to in that paper as the "intermediate" configuration) using a multi-dimensional parameter search pertinent to ice hydrometeors, i.e., cloud overlap, convective water mixing ratio, PSD, and hydrometeor types for snow, graupel, and cloud ice. The configurations for rain and cloud water were not updated and follow the long-standing configurations for all-sky assimilation at ECMWF (Geer and Baordo, 2014). The search of the "optimal" microphysical setup was treated as a cost minimization problem between actual observations and simulated observations from the Special Sensor Microwave Imager/Sounder (SSMIS). The analysis was conducted by means of latitude-longitude bins covering a frequency range of $\approx 19.0$–$190.0$ GHz. The selection of the

**Table 1.** Microphysical setup for all the hydrometeors: PM denotes the particle model, PSD is the particle size distribution, ciw stands for cloud ice water, and clw is the cloud liquid water. $D_{\min}$ and $D_{\max}$ denote the hydrometeor minimum and maximum sizes of the maximum diameter, while $a$ and $b$ comprise the coefficients of the mass-size relation that links the hydrometeor mass ($m$) to its size (maximum or geometric diameter; $D$), i.e., $m = a \cdot D^b$.

| Type | PM | $D_{\min}$ [m] | $D_{\max}$ [m] | $a$ | $b$ | PSD |
|------|-----|------------|------------|-----|-----|-----|
| rain | Mie sphere | 1.00e-4 | 1.00e-2 | 523.6 | 3.00 | Marshall and Palmer (1948) |
| snow | Sector snowflake[a] (Eriksson et al., 2018) | 1.00e-4 | 1.02e-2 | 0.00082 | 1.44 | Field et al. (2007) tropical |
| graupel | 3-Bullet rosette (Liu, 2008) | 1.00e-4 | 1.00e-2 | 0.32 | 2.37 | Field et al. (2007) tropical |
| clw | Mie sphere | 5.00e-6 | 1.00e-4 | 523.6 | 3.00 | Gamma (see, Geer and Baordo, 2014) |
| ciw | Large column aggregate[b] (Eriksson et al., 2018) | 2.42e-5 | 2.00e-2 | 0.28 | 2.44 | Heymsfield et al. (2013) stratiform |

[a]In Eriksson et al. (2018), the minimum available size of the sector snowflake is 2.00e-5 m, but following Field et al. (2007), a 1.00e-4 m size cutoff has been applied.

[b]The large column aggregate is a mixture of two hydrometeors. Below a size of 3.68e-04 m, it is complemented with the long column single crystals (Eriksson et al., 2018) to provide a complete coverage in size. The name of the mixed hydrometeor follows the one comprising the majority range of size.

hydrometeor type was made on the basis of their bulk scattering properties, with the latitude-longitude binning enhancing the spatial differentiation between the three ice hydrometeor types, and the frequency range supporting the latter differentiation owing to the explicit spectral variation of the bulk properties. Each hydrometeor type is parameterized by different $a$ and

$b$ coefficients that link the hydrometeor mass ($m$) to its size (maximum or geometric diameter; $D$), i.e., mass-size relation ($m = a \cdot D^b$). Accordingly, each hydrometeor leads to a distinct shape of the PSD (e.g., Eriksson et al., 2015) and, consequently, distinct bulk scattering "signature". Although, the morphology of the selected hydrometeors may not be considered as the most physically correct representations, it is only the bulk scattering signature that needs to be correct in the context of DA.

    The search yielded the following microphysical representation (see Table 1), which was an update on a configuration found

by Geer and Baordo (2014). The PSD introduced by Field et al. (2007) (tropical configuration) was retained as a good representation for snow within the context of DA (e.g., Geer and Baordo, 2014; Fox, 2020) and, was similarly found as a good option to represent graupel in the IFS. Snow was previously represented by the sector snowflake by Liu (2008), but it was found that results could be improved with either the ARTS sector snowflake or the ARTS large plate aggregate. Note that the ARTS sector has near identical single scattering properties to Liu's one, but it has less bulk scattering driven by a smaller $a$ coeffi-

cient of the mass-size relation. The ARTS large plate aggregate, even though it is characterized by different single scattering properties and mass-size relation, gives (for the chosen PSD) similar bulk scattering properties. This is an illustration why the particle morphology is not yet fully constrained. The choice of the ARTS sector was on the basis that it gave a slightly better fit between observations from SSMIS and the equivalent simulated observations by the IFS. For the representation of graupel, similar considerations, particularly to achieve a bulk scattering signature with weak scattering at low frequencies ($\approx 50.0$ GHz)

and strong scattering at high frequencies ($\approx 190.0$ GHz) led to the selection of the Liu 3-bullet rosette. For cloud ice, the ARTS large column aggregate (LCA) is chosen to replace the physically unrealistic Mie sphere (Geer and Baordo, 2014). Here, the choice of PSD was the main issue, since most of the available PSDs (i.e. McFarquhar and Heymsfield, 1997; Field et al., 2007) were found to generate many large (cm in size) hydrometeors, inducing too much scattering at the highest frequencies consid-

ered and leading to rather large discrepancies between observations and simulations. The Heymsfield et al. (2013) PSD was
hence chosen as it generates fewer large-size particles. Further, it is constructed on the basis of up-to-date aircraft observations
of ice containing clouds (stratiform configuration).

Note here that the polarized scattering correction scheme that is introduced later in Sect. 2.5 was used by Geer (2021) in
order to derive a final and fully consistent microphysical configuration for RTTOV v13.0.

## 2.3 Assimilation system

The ECMWF forecasting system employs a semi-Langrangian atmospheric model and a 12 h cycling four dimensional (4D)
variational DA (Rabier et al., 2000) in order to generate global analyses and 10 d forecasts. In this system, the direct assimila-
tion of all-sky radiances from MW imagers went operational in 2009 and was subsequently extended to other sensors including
microwave humidity sounders (Geer et al., 2017). Among the other data assimilated are in-situ conventional measurements,
radiances from polar orbiting satellites, e.g., Advanced Microwave Sounding Unit-A (AMSU-A), Infrared Atmospheric Sound-
ing Interferometer (IASI), Advanced Technology Microwave Sounder (ATMS), geostationary radiances, satellite-derived at-
mospheric motion vectors, and surface winds from scatterometers. Note here that microwave imager observations are averaged
("superobbed") in boxes of about 80 km by 80 km to meet the effective resolution in the model (Geer and Bauer, 2010).

The assimilation system is driven by the various observations via the background departure (d) (Geer and Baordo, 2014):

$$d = y^{\mathrm{o}} - y^{\mathrm{b}}. \tag{2}$$

Where $y^{\mathrm{o}}$ and $y^{\mathrm{b}}$ are the actual observations and the equivalent observations simulated from the background forecast:

$$y^{\mathrm{b}} = H[M[x^{\mathrm{b}}(t_0)]] + c. \tag{3}$$

The background atmospheric state, $x^{\mathrm{b}}(t_0)$, is a 12 h forecast from an earlier analysis, with $t_0$ representing the start of the new
assimilation window (Geer and Baordo, 2014). The nonlinear forecast model, $M$, propagates the atmospheric state from $x^{\mathrm{b}}(t_0)$
to the time of the observation and the forward operator $H$ (herein, RTTOV-SCATT) simulates the satellite-observed radiance
from the model profile interpolated to the observation location. Finally, $c$ denotes the bias correction term that accounts for
systematic deviations between observations and simulations, which are modelled as a linear function of total column water
vapour, surface wind speed, satellite scan angle, and other predictors; this is estimated as part of the DA process (Dee, 2004;
Auligné et al., 2007).

The ECMWF assimilation model includes four prognostic hydrometeors, i.e., rain, snow, cloud water and ice, to represent
the large-scale cloud processes, along with prognostic cloud and diagnostic precipitation fractions (Tiedtke, 1993; Forbes
et al., 2011). Convective rain and snow are diagnostic variables, derived from a mass flux convection scheme that assumes the
convective cores occupy only 5 % of each grid box. When forming the input to RTTOV-SCATT, convective rain is added to the
large-scale counterpart, but the convective snow is treated as a separate graupel hydrometeor type as explained above.

**Table 2.** Channel characteristics of the Advanced Microwave Scanning Radiometer-2 (AMSR-2), Global Precipitation Mission microwave imager (GMI), and the Special Sensor Microwave Imager/Sounder (SSMIS) considered in this study.

| Frequency [GHz] | Polarization | Instruments and No. | | |
|---|---|---|---|---|
| | | ASMR-2 | GMI | SSMIS |
| 18.7 | H | 7 | 3 | - |
| 18.7 | V | 8 | 4 | - |
| 19.35 | H | - | - | 12 |
| 19.35 | V | - | - | 13 |
| 22.235 | V | - | - | 14 |
| 23.8 | H | 9 | - | - |
| 23.8 | V | 10 | 5 | - |
| 36.5 | V | 11 | 6 | - |
| 37.0 | V | - | - | 16 |
| 89.0 | H | - | - | - |
| 89.0 | V | 13 | 8 | - |
| 91.655 | V | - | - | 17 |
| 150.0 | H | - | - | 8 |
| 166.5 | H | - | 11 | - |
| 166.5 | V | - | 10 | - |
| 183.31±7.0 | V | - | 13 | - |
| 183.31±6.6 | H | - | - | 9 |
| 183.31±3.0 | H | - | - | 10 |
| 183.31±3.0 | V | - | 12 | - |
| 183.31±1.0 | H | - | - | 11 |

## 2.4 Observations

This study combines observations from GMI, the Advanced Microwave Scanning Radiometer-2 (AMSR-2), and SSMIS. Table 2 summarizes the channels of these instruments that are actively assimilated in the cycling DA experiments.

### 2.4.1 Advanced Microwave Scanning Radiometer-2

The AMSR-2 is a conical scanning instrument with 16 channels ranging in frequency between 6.9 and 89.0 GHz, with dual-polarization capabilities. AMSR-2, flies on board the Japan Aerospace Exploration Agency's (JAXA) Global Change Observa-
180 tion Mission 1st-Water (GCOM-W1) satellite and supplies global observations at a high spatial resolution and an earth incident angle of 55 ° (Du et al., 2017).

### 2.4.2 Global Precipitation Mission microwave imager

The GPM core satellite carries a conical scanning MW radiometer, i.e., the GMI. It operates at frequencies between 10.65 and 190.31 GHz (13 channels overall) at a rather high spatial resolution, i.e., between $\approx 26$ and $\approx 6\,\text{km}$, depending on the channel. The earth incident angle is 52.8 ° for channels 1–9 and 49.1 ° for channels 10–13. GMI is the only operational passive instrument with dual polarization channels at high frequencies, i.e., 166.5 GHz, offering unique capabilities in sounding ice hydrometeors, among others.

### 2.4.3 Special Sensor Microwave Imager/Sounder

The SSMIS is a conical MW instrument, with an earth incident angle of 53.1 °, which is on board the Defense Meteorological Satellite Program F-17 spacecraft (Sun and Weng, 2012). It consists of 24 channels between $\approx 19.0$ and $\approx 190.0\,\text{GHz}$ (21 frequencies in total), three out of which are in both V- and H-polarization, i.e., 19.35, 37.0, and $\approx 91.65\,\text{GHz}$. Compared to AMSR-2 and GMI, SSMIS has a lower spatial resolution, with an instantaneous field of view ranging from $\approx 54\,\text{km}$ (at the lowest frequency channels) to $\approx 14\,\text{km}$ (at the highest frequency channels).

## 2.5 Polarized scattering correction

A widely used simplification in radiative transfer modeling is the assumption of totally randomly oriented (TRO) hydrometeors. Such hydrometeors are considered as macroscopically isotropic, meaning that there is no favored propagation direction or orientation and any induced polarization signal is only driven by the radiation scattered into the line of sight (e.g., see Barlakas, 2016). However, ice hydrometeors are characterized by non-spherical shapes and thus non-unit aspect ratios. This could potentially lead to preferential orientation driven by gravitational and aerodynamical forces (Khvorostyanov and Curry, 2014) or even by electrification processes (lightning activities in deep convective systems, Prigent et al., 2005). Under turbulence-free conditions, small non-spherical hydrometeors (diameters below $\approx 10\,\mu\text{m}$) are totally randomly oriented owing to Brownian motion (Klett, 1995); but, if they are large enough, they tend to be horizontally oriented as they fall depending on their shape: this holds true for thick plates with a diameter above $\approx 40\,\mu\text{m}$ (Klett, 1995), while oblate spheroids and thin plates would adopt horizontal orientation at sizes larger than $\approx 100\,\mu\text{m}$ (Prigent et al., 2005, and references therein) and $\approx 150\,\mu\text{m}$ (Noel and Sassen, 2005), respectively. However, turbulent effects can easily disrupt any orientation especially for small hydrometeors or introduce a wobbling motion around the horizontal plane at larger sizes (10–30 $\mu\text{m}$) (Klett, 1995). In addition, tumbling motions in strong turbulent conditions, e.g., within deep convective cores, induce total random orientation (Spencer et al., 1989).

Hydrometeor orientation results in anisotropic scattering with viewing-dependent optical properties, meaning different values of extinction at different polarization components of the incident radiation (dichroism effect, Davis et al., 2005). Accordingly, in vector radiative transfer theory, the attenuation between the incident radiation and the hydrometeor is governed by a 4 x 4 extinction matrix **K**, depending on the incident direction and the orientation of the hydrometeor with respect to a reference system (e.g., Barlakas, 2016). This reference system, or, in other words, the laboratory system is a three-dimensional Cartesian

coordinate one that is characterized by a specified position in space and sharing the same origin as the hydrometeor coordinate

system; note that for typical applications, the $z$ axis of the laboratory system would be aligned with the local vertical direction. Although the selection of the hydrometeor system is arbitrary, it is commonly specified by the shape of the hydrometeor (Mishchenko et al., 1999). In case of an axial symmetry, for example, it is common to align the hydrometeor $z$ axis along the direction of symmetry, with its largest dimension being aligned to the $x$–$y$ plane (Brath et al., 2020). This acknowledges the tendency for hydrometeors to have horizontal alignment. Now, the orientation of the hydrometeor coordinate system, i.e., hydrometeor orientation, in regard to the laboratory system can be described by the three Euler rotation angles (rotations are applied in order): $\alpha \in [0, 2\pi]$ around the laboratory $z$ axis, $\beta \in [0, \pi]$ around the hydrometeor $y$ axis, and $\gamma \in [0, 2\pi]$ around the hydrometeor $z$ axis (see Fig. 1 in Brath et al., 2020).

In practice, scattering media consists of an ensemble of hydrometeors with various orientations. Thus, the single scattering properties should be averaged over all possible orientations to derive the corresponding scattering properties of an ensemble of oriented hydrometeors (Mishchenko and Yurkin, 2017):

$$\langle f \rangle = \int\limits_{0}^{2\pi} \int\limits_{0}^{\pi} \int\limits_{0}^{2\pi} p(\alpha) \cdot p(\beta) \cdot p(\gamma) \cdot f \, \mathrm{d}\alpha \, \mathrm{d}\beta \, \mathrm{d}\gamma. \tag{4}$$

Where $f$ is a single scattering property (e.g., extinction matrix) at a specific orientation, while $p(\alpha)$, $p(\beta)$, and $p(\gamma)$ describe the probability distributions of the three Euler angles.

In case of TRO, all possible orientations are equally likely to occur and $p(\alpha)$, $p(\beta)$, and $p(\gamma)$ describe uniform distributions. Consequently, the extinction matrix has no angular dependency and it is reduced to its first element $K_{\mathrm{TRO}} = K_{11}$, which describes the extinction cross-section (Mishchenko and Yurkin, 2017; Brath et al., 2020). However, gravitational and/or aerodynamical forces can induce an axial symmetry, with the axis of symmetry specified by the direction of the force (Mishchenko et al., 1999). By aligning the laboratory $z$ axis along the direction of the force, $p(\alpha)$ and $p(\gamma)$ become uniform distributions resulting in an axial symmetry depending on $\beta$, i.e., tilt angle (Mishchenko et al., 1999). This results in the so-called azimuth random orientation (ARO), describing a preferred orientation to the horizon based on the tilt angle, with no favored orientation in the azimuth direction (Brath et al., 2020). In this case, **K** depends on the incident direction and the tilt angle and it is reduced to only three independent elements ($K_{11}$, $K_{12}$, and $K_{34}$). $K_{12}$ represents the differences in the extinction between V- and H-polarization (cross section for linear polarization), while $K_{34}$ describes the differences in the extinction between $+45\,°$ and $-45\,°$ polarization (cross section for circular polarization), and is not relevant here. For a comprehensive description of ARO, the reader is referred to Brath et al. (2020).

Figure 1 highlights the differences in **K** between ARO and TRO for the large plate aggregate (LPA) at 166.9 GHz. It displays the first two elements of **K** normalised by the extinction at TRO as a function of the incident angle ($\theta_{\mathrm{inc}}$), for four tilt angles, and two values of the size parameter $x$:

$$x = \pi \cdot D_{\mathrm{veq}} / \lambda, \tag{5}$$

where $D_{\mathrm{veq}}$ is the volume-equivalent diameter of the hydrometeor and $\lambda$ is the wavelength. In case of ARO, the scattering data of Brath et al. (2020) has been utilized. At the tilt angles considered here (0–60 °), hydrometeor orientation induces differences

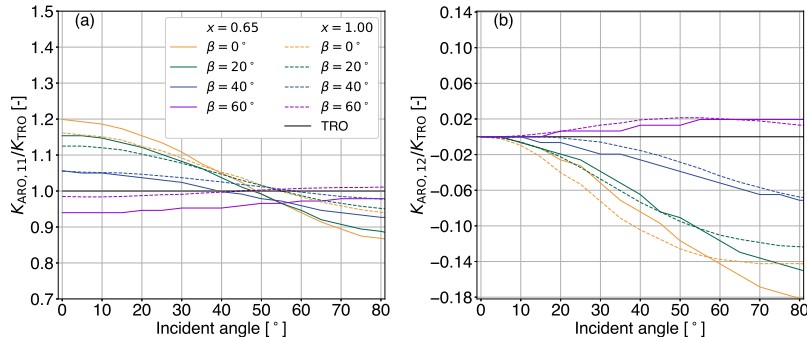

**Figure 1.** For large plate aggregate, extinction matrix elements of azimuth random orientation (ARO) normalized by the extinction cross section in case of the total random orientation (TRO) as a function of incident angle $\theta_{\mathrm{inc}}$ at 166.9 GHz. Results are presented in case of (a) $K_{\mathrm{ARO,11}}/K_{\mathrm{TRO}}$ and (b) $K_{\mathrm{ARO,12}}/K_{\mathrm{TRO}}$ for two size parameters ($x = 0.65$ in solid lines and $x = 1.00$ in dashed lines) and four tilt angles $\beta$ highlighted by the different colours. In black, the values in case of TRO are depicted.

up to 20 % of $K_{\mathrm{ARO,11}}$ compared to $K_{\mathrm{TRO}}$. The maximum differences are reported at $\theta_{\mathrm{inc}}$ of $0\,^\circ$, while the smallest ones are found at around $\theta_{\mathrm{inc}}$ of $55\,^\circ$ and are in the range of $\approx 0$–$4\,\%$ (depending on $x$). In fact, for this incidence angle, the differences between $K_{\mathrm{ARO,11}}$ and $K_{\mathrm{TRO}}$ are close to zero over all sizes (not shown here). This implies that at earth incident angles around 250    $55\,^\circ$ (observation angle of conical scanners), hydrometeor orientation does not change the overall level of extinction.

At small $\theta_{\mathrm{inc}}$, the cross section for linear polarization ($K_{\mathrm{ARO,12}}$) is close to zero, meaning that the differences in the extinction between V- and H-polarization are negligible. However, these differences are increasing with increasing $\theta_{\mathrm{inc}}$ up to about $\approx 80\,^\circ$. The largest differences in the extinction between V- and H-polarization (up to about 18 %) are derived for a tilt angle of $0\,^\circ$. Recall here that results are presented in case of the LPA shape. For other shapes, e.g., plate type, the magnitude of 255    $K_{\mathrm{ARO,12}}$ is much larger (Brath et al., 2020). In any case, there are large differences in the cross section for linear polarization ($K_{12}$) between ARO and TRO hydrometeors, even at earth incident angles around $55\,^\circ$. This (rather than any change in $K_{11}$) is the likely explanation for the polarized scattering observed by microwave imagers.

To that end, a simple scheme has been implemented into RTTOV-SCATT to improve the physical representation of polarized scattering. To model the effect of ARO ice hydrometeors (snow, graupel, and cloud ice) in recreating the observed polarization 260    signal from conical scanning radiometers, the layer optical thickness of TRO hydrometeors ($\tau$) is increased in H-polarized channels and decreased in V-polarized channels by a correction factor $\alpha$, leading to the polarization ratio ($\rho$):

$$\rho = \frac{\tau_{\mathrm{H}}}{\tau_{\mathrm{V}}} = \frac{\tau \cdot (1 + \alpha)}{\tau \cdot (1 - \alpha)} = \frac{1 + \alpha}{1 - \alpha}. \tag{6}$$

Where $\tau_{\mathrm{H}}$ and $\tau_{\mathrm{V}}$ are the corrected optical thickness at H- and V-channels, respectively. This ratio can be seen as an indirect representation of the axial ratio of an oriented hydrometeor, while $\alpha$ approximates the differences in the extinction between the two channels ($K_{\mathrm{ARO,12}}$; see Fig. 1b). In other words, $\alpha$ forces the extinction at V-polarization (H-polarization) to be weaker

(stronger). Note here that the asymmetry parameter and the single scattering albedo have been not modified, as it is not yet clear how to do so.

## 2.6 Experiments

Data assimilation experiments were conducted utilizing the 46r1 cycle of the IFS of ECMWF. This cycle was operational from June 2019 to June 2020. Compared to the operational system, the microphysical configuration of RTTOV-SCATT has been upgraded as described earlier, and a slightly reduced resolution of about 28 km (labeled Tco399) with 137 vertical levels has been used. Two types of experiments are considered:

1. To scrutinize the performance of the forward operator, a consistent sample of observations should be examined. Consequently, passive monitoring experiments were conducted to simulate GMI radiances. In the monitoring mode, the short-term forecast for all experiments is supplied from a single parent cycling DA experiment. In other words, the background forecasts (sometimes referred to as first guess) comes from the parent experiment and changes to the usage of GMI observations do not feed back; the background atmospheric state is kept constant. Nine experiments have been conducted for a period of one month (13 June to 13 July 2019), with $\rho$ ranging from 1.1 to 1.5, with an increment of 0.05. Additional experiments were run in which the $\rho$ was applied to each ice hydrometeor individually; for details the reader is referred to Sect. 3.3.

2. To assess the potential long term impact of $\rho$ on the forecast, cycled DA experiments were run for a total period of 6 months, i.e., from 15 February to 31 May 2019 and from 13 June to 31 August 2019. Here, the improved forward operator is used for AMSR-2, GMI, and SSMIS as part of the DA process. The resulting analysis provides the initial conditions for the background forecast in the next cycle of DA. Hence, in contrast to the passive experiments, the background forecast and observational usage vary, and the changes can feed back over multiple cycles of DA. These experiments are described further in Sect. 3.4.

For the passive monitoring experiments, the period of one month was chosen on the basis that polarisation signatures of oriented ice hydrometeors are fairly consistent through the seasons and across the globe (Gong and Wu, 2017; Galligani et al., 2021). For both types of experiments, to highlight whether the forward operator and the forecast are improved or degraded, control experiments were additionally run, with the improved physical representation of polarized scattering being turned off, meaning a $\rho$ value of 1.0 (a correction factor of 0.0).

## 3 Choosing the best polarization ratio

## 3.1 Overview

By means of passive monitoring experiments, the ability of $\rho$ to simulate the observed polarization patterns due to oriented ice hydrometeors is explored. Accordingly, a special emphasis is given on the high frequency dual-polarization channels of

GMI (89.0 GHz and especially on 166.5 GHz), because scattering due to ice hydrometeors (snow, graupel, and cloud ice) is increasing with frequency. For clarity, we discriminate between the observed (o) and simulated (background, b) polarization difference:

$$\mathrm{PD}^{\mathrm{o}} = T_{\mathrm{BV}}^{\mathrm{o}} - T_{\mathrm{BH}}^{\mathrm{o}},$$

$$\mathrm{PD}^{\mathrm{b}} = T_{\mathrm{BV}}^{\mathrm{b}} - T_{\mathrm{BH}}^{\mathrm{b}}. \tag{7}$$

One of the challenges encountered was to screen out any surface contamination and particularly, any strong polarization signal that originates from water surfaces (Meissner and Wentz, 2012); strongly polarized surfaces complicate the separation between cloudy and clear sky conditions. Appendix A describes a number of careful screening checks that were used to minimize the surface contribution. Note here that throughout Sect. 3, results are presented in terms of the screened data that are almost entirely free from surface contribution.

Figure 2 illustrates the global PD as a function of $T_{\mathrm{BV}}$ at 166.5 GHz in case of both observations and simulations. Polarized scattering from preferentially oriented ice hydrometeors leads to $\mathrm{PD}^{\mathrm{o}}$ up to 10–15 K (in red) centered around 220 K, with increasingly low $T_{\mathrm{BV}}$ indicating increasingly cloud-affected scenes. In fact, the arch-like shape of the $\mathrm{PD}^{\mathrm{o}} - T_{\mathrm{BV}}^{\mathrm{o}}$ (or $T_{\mathrm{BH}}^{\mathrm{o}}$) relation appears to be universal (Gong and Wu, 2017). The existing modelling framework characterized by a polarization ratio of 1.0 (control run; in green) completely fails to reproduce such polarization signal (see Fig. 2a). However, it provides valuable information regarding the surface contribution; if there was any simulated polarization signal due to ocean reflection, it would be visible. The remaining panels in Fig.2, i.e., b–f, depict the ability of $\rho$ to provide realistic simulations of this behaviour, and a first glance indicates that a $\rho$ value between 1.2 and 1.4 could do a reasonable job, since it is within the distributions of the observations.

## 3.2 Quantifying the fit of model to observations

In order to quantify the best fit between simulations and observations, the commonly used metrics are the mean, the standard deviation ($\sigma$), and/or the root mean square error (rmse). Nonetheless, forecast modelling systems are still unable to predict cloud and precipitation at small scales, and thus, are characterized by mislocation errors that could lead to quite large $\sigma$ and/or rmse (e.g., Geer and Baordo, 2014, and references therein). A more comprehensive assessment of the polarized scattering correction is achieved by measuring the divergence between the distributions of observations and simulations introduced in Geer and Baordo (2014):

$$\Phi = \frac{\left(\sum_{\mathrm{bins}} |\varphi|\right)}{\#\mathrm{bins}} = \left(\sum_{\mathrm{bins}} \left| \log_{10} \frac{\#P^{\mathrm{b}}}{\#P^{\mathrm{o}}} \right| \right) / (\#\mathrm{bins}), \tag{8}$$

where $P^{\mathrm{b}}$ and $P^{\mathrm{o}}$ are the populations of simulations and observations, respectively, in each bin, while $\#\mathrm{bins}$ denotes the total number of bins used in the comparison (which is a minor modification from the original approach). The metric of divergence describes how well the experiments approximate the observations by putting a penalty according to the logarithm base ten of the ratio of the populations ($\varphi$; simulations divided by observations) in each bin. To avoid any infinite penalty in case of zero

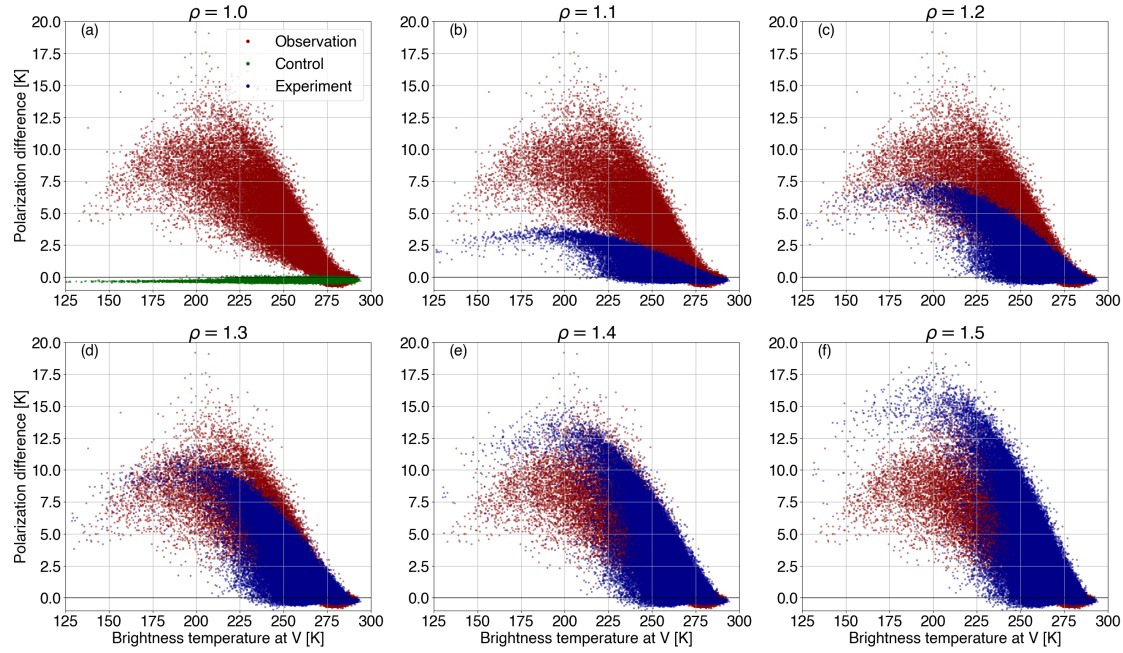

**Figure 2.** Polarization difference between the vertically and horizontally polarized channels (PD $= T_{\mathrm{BV}} - T_{\mathrm{BH}}$) as a function of $T_{\mathrm{BV}}$ at 166.5 GHz as observed by GMI (PD$^{\mathrm{o}}$; in red) and simulated (PD$^{\mathrm{b}}$; in blue) by the passive monitoring experiments for a period of one month (13 June to 13 July 2019). Results are presented for polarization ratios ($\rho$) ranging from 1.0 (control run; in green) to 1.5, with a 0.1 increment.

populations, empty bins are replaced by a population of 0.1; varying this limit has a minor effect and increases or decreases the penalty effect. The complete set of statistics used here are the mean and skewness of PD$^{\mathrm{o}}$ − PD$^{\mathrm{b}}$, the corresponding histogram divergence, i.e., one dimensional (1D) divergence, as well as the two dimensional (2D) divergence between the observed (PD$^{\mathrm{o}}$ − $T_{\mathrm{BV}}^{\mathrm{o}}$) and simulated (PD$^{\mathrm{b}}$ − $T_{\mathrm{BV}}^{\mathrm{b}}$) 2D histograms of the arch-like relationship, which are non-localised measures of the discrepancy between simulated and observed distributions.

Figure 3 depicts these statistical metrics with respect to the polarization ratio. Results are reported globally (in black crosses), but the situations over ocean (in blue triangles) and land (in brown dots) are also described. Note here that in case of ocean and land, the pixels across the coastline are excluded, while "global" represents the overall number of pixels (including the coastline). To begin with, the control run, which is highlighted by the corresponding green markers, is unable to provide realistic simulations of the observed PDs at both 89.0 (upper panels) and 166.5 GHz (lower panels); it leads to a mean error of about 1 K (globally) and the histograms (of PD$^{\mathrm{o}}$ − PD$^{\mathrm{b}}$, not shown here) are described by right-skewed distributions that are more asymmetric over the land compared to the ocean (see skewness; second panel at each row). In both frequencies, even a small increase to the polarization ratio above 1.0 reduces discrepancies (mean error; first panel at each row) between observations and simulations and leads to a more symmetrical error (skewness; second panel at each row); further increasing

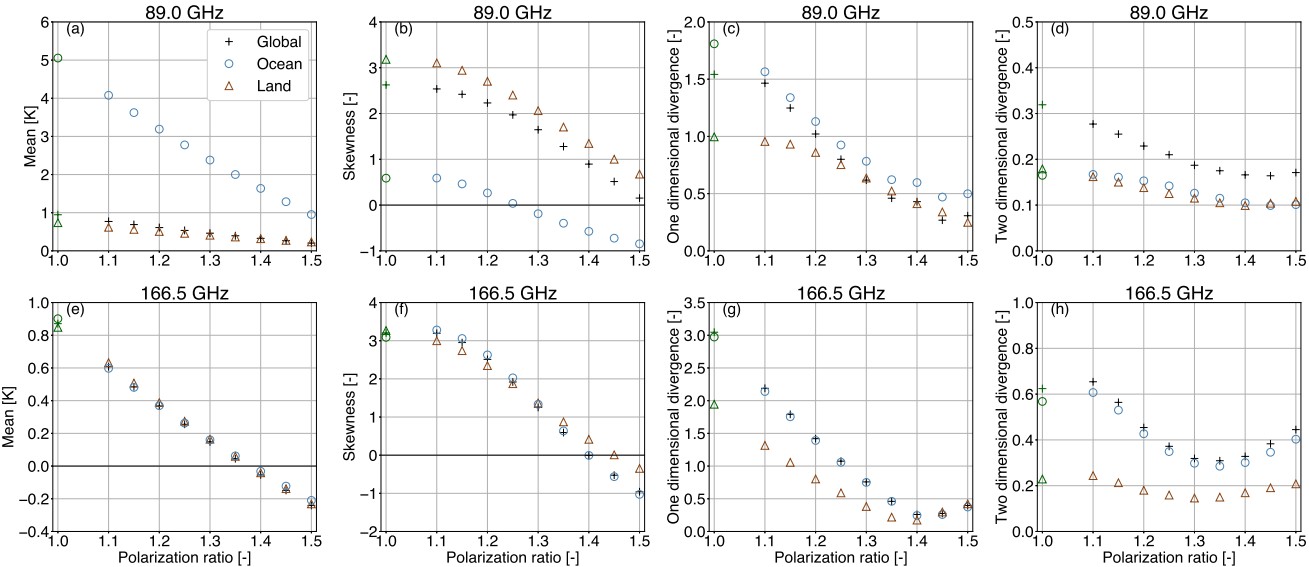

**Figure 3.** Statistical metrics, i.e., mean, skewness, and one dimensional (1D) divergence describing the differences in polarization differences between observations and simulations, i.e., $PD^o - PD^b$, and the two dimensional (2D) divergence between the observed ($PD^o - T^o_{BV}$) and simulated ($PD^b - T^b_{BV}$) 2D histograms of the arch-like relationship. Results are presented for 89.0 GHz (top) and 166.5 GHz (bottom) over land (in brown triangles), ocean (in blue circles), and globally (in black crosses) for a period of one month (13 June to 13 July 2019) as a function of the polarization ratio ($\rho$). In terms of the control run ($\rho = 1$), the corresponding differences are highlighted in green.

polarization ratio reduces both the mean and the skewness of $PD^o - PD^b$ still further up to a point where the best ratio can be found. At 89.0 GHz and over the entire globe, the best polarization ratio is 1.5 (skewness is close to zero). However, over the ocean, a more symmetrical error is achieved by a smaller value, i.e., 1.45, but this could be attributed to the relatively low number of pixels that passed through the rather strict screening method that minimizes the strongly polarizing ocean surface.

On the contrary, the dependency of the 166.5 GHz channel on the polarization ratio is characterized by similar behaviour over both land and ocean. An increase of $\rho$ improves the representation of preferentially oriented ice hydrometeors up to a value of about 1.4 (in terms of the mean and the skewness); a further increase of $\rho$, increases $PD^o - PD^b$ once again, but towards negative values. This implies that the corresponding histogram, from a right-skewed distribution, becomes symmetrical at a polarization ratio of 1.4 and turns into a left-skewed one at higher $\rho$ values.

In Fig. 3, the overall 1D and 2D divergence - $\Phi^{1D}$ (third panel at each row) and $\Phi^{2D}$ (forth panel at each row), respectively - derived by Eq. (8) are also shown. At 166.5 GHz, the 1D divergence is in line with the conclusions drawn from the mean and skewness, while the corresponding 2D divergence marginally suggests that the minimum value is found at the lower polarization ratio of 1.35. At 89.0 GHz, both the 1D and 2D global divergence marginally pinpoint that the minimum value is acquired for the lower polarization ratio of 1.45, but results for $1.4 < \rho < 1.5$ present low sensitivity to the selected polarization

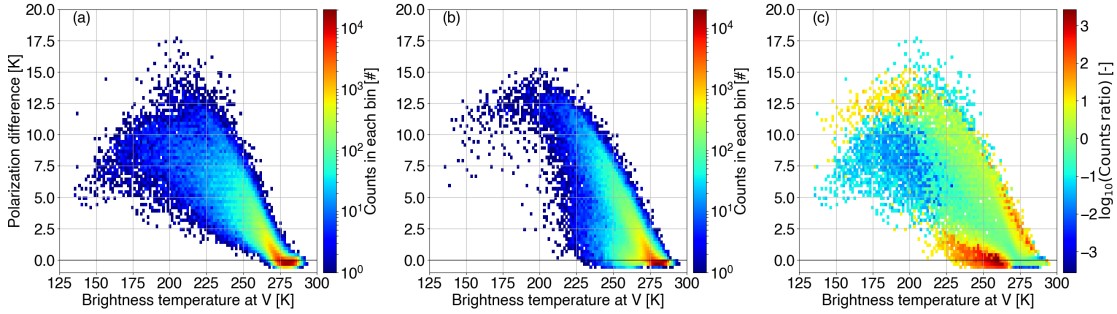

**Figure 4.** Two dimensional (2D) histograms describing the arch-like relationship between the polarization difference and the brightness temperature at V-polarization at 166.5 GHz as (a) observed ($\mathrm{PD^o} - T_{\mathrm{BV}}^{\mathrm{o}}$) by GMI, (b) simulated ($\mathrm{PD^b} - T_{\mathrm{BV}}^{\mathrm{b}}$) for a polarization ratio of 1.4, and (c) the 2D histogram divergence between $\mathrm{PD^o} - T_{\mathrm{BV}}^{\mathrm{o}}$ and $\mathrm{PD^b} - T_{\mathrm{BV}}^{\mathrm{b}}$. In panel (c), white areas denote the case where both the observed and the simulated 2D bins are empty.

ratio; especially in terms of the $\Phi^{2\mathrm{D}}$. However, data assimilation assumes that errors are Gaussian and unbiased; hence, we prioritize minimising the measure of skewness, rather than the 2D divergence.

To conclude, the polarization ratio that models best the orientation of non-spherical ice hydrometeors and leads to a more symmetrical error within the IFS is 1.5 for 89.0 GHz and 1.4 for 166.5 GHz. This is translated to increasing/decreasing $\tau_{\mathrm{H}}$ ($\tau_{\mathrm{V}}$) by 20 % ($a = 0.2$) at 89.0 GHz and by 16.7 % ($a = 0.167$) at 166.5 GHz. Note here that there is low confidence in the optimal value of $\rho$ obtained at 89.0 GHz; clarifications are given in Sect. 4.2. Consequently, the polarization ratio found at 166.5 GHz is chosen to further assess the impact of the polarized scheme on both the forward operator and the forecast impact.

Figure 4 provides an overview of the global performance of the polarization ratio of 1.4 to simulate the observed PDs at 166.5 GHz. To begin with, Fig. 4a illustrates the 2D histogram describing the observed arch-like relationship of $\mathrm{PD^o} - T_{\mathrm{BV}}^{\mathrm{o}}$ introduced in Fig. 2 (in red). Most of the points are bundled above $\approx 275$ K, which are linked to clear-sky conditions and PDs close to 0 K. The cloud induced PDs are mostly accumulated at $T_{\mathrm{BV}}^{\mathrm{o}}$ between $\approx 215$ and $\approx 275$ K. Figure 4b depicts the corresponding simulated $\mathrm{PD^b} - T_{\mathrm{BV}}^{\mathrm{b}}$ relation for a ratio of 1.4, while Fig. 4c shows the 2D histogram divergence between $\mathrm{PD^o} - T_{\mathrm{BV}}^{\mathrm{o}}$ and $\mathrm{PD^b} - T_{\mathrm{BV}}^{\mathrm{b}}$ highlighting the performance of the selected $\rho$; note here that the 2D bin divergence denotes the lowercase letter "$\varphi$" ($\varphi^{2\mathrm{D}}$) in Eq. (8) and not the overall divergence which is represented by the capital letter "$\Phi$" ($\Phi^{2\mathrm{D}}$). One can see that such a $\rho$ value approximates quite well the observed PDs (high slope where most of the points are accumulated), while it underperforms at values of brightness temperature below $\approx 225$ K and at the lower part of the arch ($225 < T_{\mathrm{BV}} < 250$ K) characterized by low PD values (0–2 K). For the situation at 89.0 GHz, the reader is referred to Fig. B1 in Appendix B.

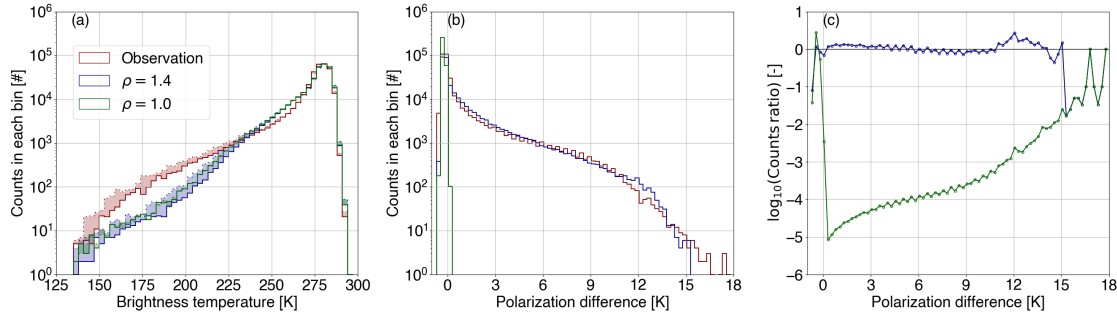

**Figure 5.** (a) Histograms of the brightness temperature at 166.5 GHz as observed ($T_B^o$; in red) by GMI and simulated ($T_B^b$) for a polarization ratio of 1.4 (in blue) and 1.0 (in green; control run); solid and dashed lines denote the brightness temperature at vertical and horizontal polarization, respectively, while the shaded area highlights their difference. (b) Corresponding histograms of polarization differences (PD) and (c) the divergence between the distributions of $PD^o$ and $PD^b$ as a function of the polarization difference.

A 1D global comparison is found in Fig. 5. Firstly, the left panel shows the histograms of the brightness temperature as observed ($T_B^o$; in red) by GMI and simulated ($T_B^b$) for a polarization ratio of 1.4 (in blue) and 1.0 (in green; control run). The solid and dashed lines denote $T_{BV}$ and $T_{BH}$, respectively, while the shaded area highlights their difference. One can see that

the IFS and RTTOV-SCATT underestimates the brightness temperature depressions linked to rather deep convective systems (at increasing low $T_B$), but a polarization ratio of 1.4 realistically represents the histogram differences between $T_{BV}$ and $T_{BH}$ (blue shaded area), that is completely missed by the control run (limited green shaded area). The middle panel illustrates the corresponding histograms of PDs. The control run (in green) completely fails to reproduce any positive PDs larger than 0.2 K (see also Fig. 2a). On the other hand, a polarization ratio of 1.4 (in blue) removes most of the discrepancies between

observations and simulations. The right panel displays the related 1D divergence ($\varphi^{1D}$), or in other words, the logarithm base ten of the ratio of the populations (see Eq. 8). The experiment with $\rho = 1.4$ slightly overestimates the occurrences of PDs between 0.2 and about 5.0 K, and thus, these are penalized with a positive logarithm base ten ratio. From around 5.0 to 10.5 K, a very good agreement is found between simulations and observations, while at negative and at the largest PD values, the largest differences are reported (this ratio does not simulate PDs > 15 K), leading to a total divergence ($\Phi^{1D}$) value of 0.260.

In terms of the control run, any $\varphi^{1D}$ value corresponding to a PD above 0.2 K results from the penalty effect assigned to avoid infinite values. The control run yields a total divergence ($\Phi^{1D}$) value of 3.047.

### 3.3 Polarization differences per hydrometeor type

To discriminate the polarization signal at 166.5 GHz induced by the various ice hydrometeors, additional passive monitoring experiments were conducted, whereby the polarization ratio of 1.4 was applied to snow, graupel, and cloud ice water indi-

vidually. Figure 6 displays a global overview of the simulated $PD^b - T_{BV}^b$ arch-like relation compared to the observed one. Preferentially oriented large scale snow produces most of the observed polarization signature, while deep convective snow

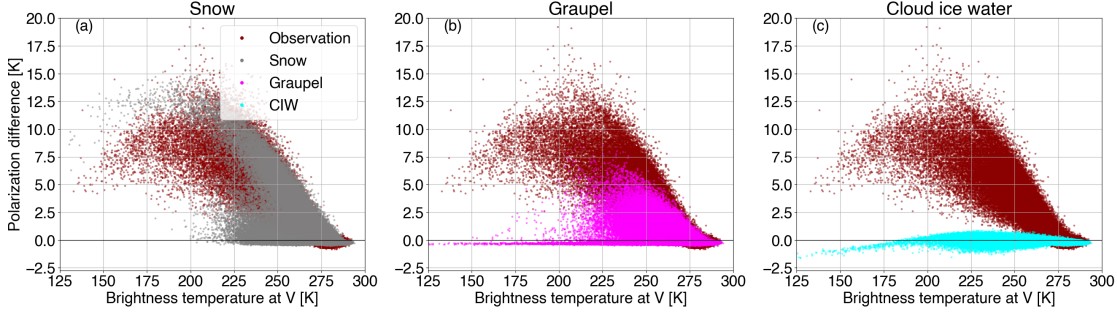

**Figure 6.** As in Fig. 2, but in case of (a) snow (in grey), (b) graupel (in magenta), and (c) cloud ice water (ciw; in cyan) individually polarized passive monitoring experiments at a polarization ratio of 1.4.

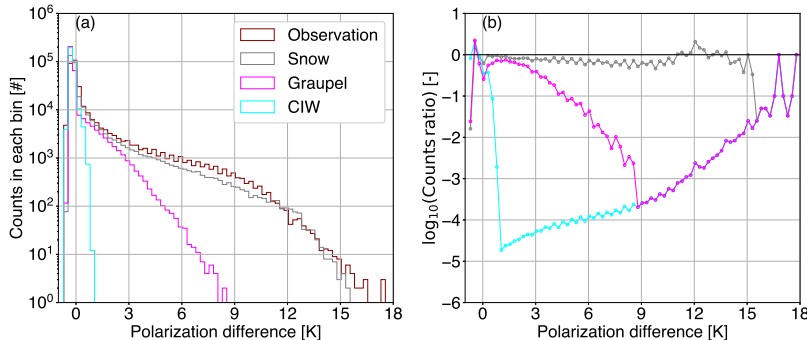

**Figure 7.** Histograms of the polarization differences at 166.5 GHz as observed by GMI (PD$^{\mathrm{o}}$; in red) and simulated (PD$^{\mathrm{b}}$) for a polarization ratio of 1.4 applied to (a) snow only (in grey), (b) graupel only (in magenta), and (c) cloud ice water only (ciw; in cyan).

(graupel) induces only low-to-medium PD values and a much lower peak ($\approx 7.5\,\mathrm{K}$). Interestingly, cloud ice produces a minor polarization signal, with negative values at increasingly cold $T_{\mathrm{BV}}$.

To quantify the contribution of each ice hydrometeor to the total polarization difference (globally), the consistency in the distributions between observations and simulations has been investigated; Figure 7 illustrates the corresponding histograms and their 1D divergence. Comparing Fig. 5b with Fig. 7a, it follows that snow slightly underestimates PDs between 2 and 11 K, leading to small negative penalty values of $\varphi^{1\mathrm{D}}$; the overall divergence $\Phi^{1\mathrm{D}}$ is found to be a bit larger (0.279) than the one resulting by applying the $\rho$ to all the hydrometeors (0.260). Graupel poorly represents the polarization differences above $\approx 2.5\,\mathrm{K}$, while it completely fails to generate any values above $\approx 8.5\,\mathrm{K}$; the underestimation is increasing with increasing PD up to $\approx 8.5\,\mathrm{K}$ and, at larger PD values, any $\varphi^{1\mathrm{D}}$ value is solely linked to the artificial penalty effect, i.e., empty bins are replaced by a population of 0.1, resulting to a total $\Phi^{1\mathrm{D}}$ value of 1.627. Cloud ice generates a polarization signal between $-0.5$ and 1.2 K, leading to a much larger total $\Phi^{1\mathrm{D}}$ (2.859); this is close to the one obtained in case of the control run, i.e., 3.047.

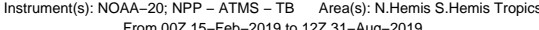

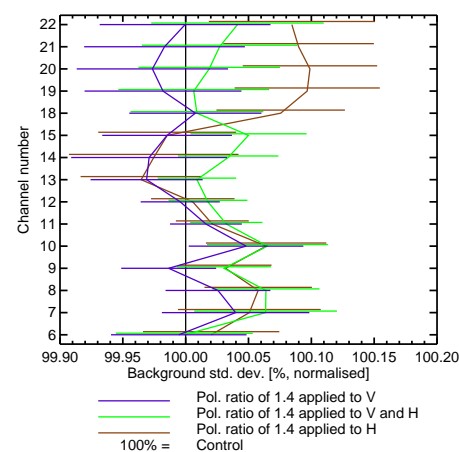

**Figure 8.** Normalised change in the standard deviation ($\sigma$) of background (12 h forecast) fits to independent ATMS observations with a polarization ratio ($\rho$) of 1.4, applied in different ways to the V- and H-channels. Error bars give the 95 % confidence interval estimated using a paired-difference t-test from per-cycle (every 12 h) statistics, without corrections for multiple comparisons.

### 3.4 Forecast impact

As described in Sec. 2.6, cycled DA experiments were run to see whether using polarized scattering in the observation operator
for SSMIS, GMI, and AMSR-2 could improve the quality of the forecast. The polarization ratio $\rho =1.4$ found at 166.5 GHz was used for all ice hydrometeors at all frequencies, but it was applied in three different ways: either with equal and opposite perturbations to V- and H-channels as in Eq. (6) where $\alpha = 0.167$, or by unilaterally changing the ice hydrometeor optical depth in only H- or only V-channels:

$$\rho = \frac{\tau_H}{\tau_V} = \frac{\tau \cdot (1+\alpha)}{\tau \cdot (1-\alpha)} = \frac{\tau \cdot (1+\beta)}{\tau \cdot (1)} = \frac{\tau \cdot (1)}{\tau \cdot (1-\gamma)} = 1.4. \tag{9}$$

These latter experiments are equivalent to increasing $\tau_H$ by 40 % ($\beta = 0.4$) or reducing $\tau_V$ by 29 % ($\gamma = 0.286$). This is an approximate way to explore whether the polarized scattering might provide better results in combination with an adjustment to the original ice hydrometeor optical depths, which are not themselves known with high accuracy.

First, we evaluate the performance of the polarized correction against independent references by using the observations and channels selected for active DA, but not yet assimilated in all-sky conditions (i.e., the observation operator is unchanged); it is
essential to ensure that improving polarized scattering for conical scanners does not degrade the data assimilation in clear sky conditions. For example, Fig. 8 examines the change in quality of the 12 h forecast as measured by the $\sigma$ of ATMS background departures. The standard deviation, $\sigma$, is an appropriate metric of forecast skill here because the distribution of simulated ATMS observations in clear-sky conditions is not expected to change. Starting with the results where $\rho =1.4$ was applied with equal and opposite perturbations to V- and H-channels for all three conical scanners, it can be seen that the ATMS tropospheric
humidity channels (18–22; around 183.31 GHz) and stratospheric channels (11–15; $\approx 56.9$–57.6 GHz) are not significantly

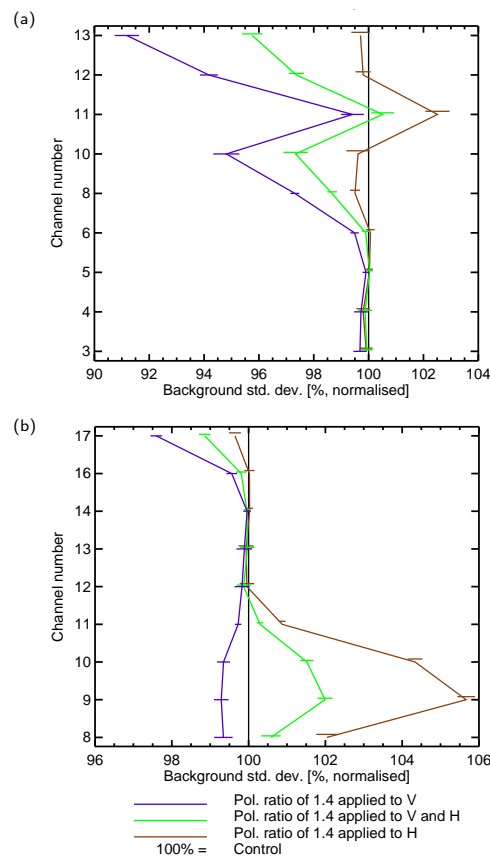

**Figure 9.** Normalised change in the standard deviation ($\sigma$) of background (12 h forecast) fits to (a) GMI and (b) SSMIS observations, with a polarization ratio ($\rho$) of 1.4, but applied in different ways to the V- and H-channels; other details as in Fig. 8. See Table 2 for channel descriptions.

affected. However the tropospheric temperature sounding channels (6–10; $\approx 53.6$–57.3 GHz) show a slight but statistically significant increase in $\sigma$, suggesting a minor degradation in forecast quality. To counterbalance this, other conventional data (not shown) showed marginal but significant improvements in forecast quality, suggesting overall a neutral impact.

Slightly different results were seen with the unilateral scaling of the layer optical thickness between V- and H-channels. The unilateral increase in H-channels appears to give a significant degradation in forecast skill in tropospheric temperature channels (6–10; $\approx 53.6$–57.3 GHz) and humidity channels (18–22; around 183.31 GHz) of ATMS. In contrast, a unilateral decrease in V-channels appears to leave forecast skill unchanged. As the overall level of hydrometeor optical depth decreases (going from H only, to V and H, to V only), it appears that the forecast skill slightly improves. This suggests that it is also important to get the overall level of scattering correct to be able to implement the polarized scattering.

Figure 9 shows the normalised change in $\sigma$ of background departures for two of the sensors for which the polarized scattering correction has been applied, i.e., GMI and SSMIS. Here, changes can come from two sources: a change in the quality of the background forecasts (as for ATMS) or directly due to the change in the observation operator (a least, in channels where the scaling was applied). In this second effect, when the observation operator is modified, statistics can be affected by the change in the distributions of simulated brightness temperatures, and hence they are vulnerable to the double penalty effect. For example,

adding 40 % to ice hydrometeor optical depths in an H-channel like SSMIS 9 (183.31±6.6 GHz; the brown line in Fig. 9b) likely explains the 6 % larger $\sigma$ of background departure. Similarly, reducing extinction by 29 % in V-channels such as SSMIS 17 (91.655 GHz; the purple line in Fig. 9b) likely explains the reduction in $\sigma$ by 2 %. Where polarization is applied with equal and opposite perturbations to the V- and H- channels (the green lines) the change in $\sigma$ hence appears to be dominated by the double penalty effect. However, the figure still provides an important result on the benefits of polarized scattering. As shown in Table

2, most of the high-frequency channels of GMI are V-polarized ($\geq 89.0$ GHz, channels 8, 10, 12, and 13) and most of those on SSMIS are H-polarized ($\geq 150.0$ GHz, channels 8, 9, 10, and 11). Where only the H-channel simulations are modified, the V-channels provide an independent measure of the forecast quality. Likewise when only V-channel simulations are modified, the H-channels are an independent reference. Adding extinction in H-channels (brown lines), i.e., GMI channel 11 (166.5 GHz) and SSMIS channels 8–11 ($\approx$ 150–183.31 GHz), reduces $\sigma$ by around 0.5 % in GMI V-channels, i.e., channel 10 (166.5 GHz)

and 12–13 ($\approx$ 183.31 GHz), and in SSMIS V-channel 17 (91.655 GHz). Similarly, reducing extinction in V-channels (purple lines) reduces $\sigma$ in SSMIS H-channels and in GMI H-channel 11 (166.5 GHz), similarly by around 0.5 %. Results from AMSR-2 (not shown) are equivalent. This suggests that representing polarized scattering makes the forward modelling more consistent across sensors, as well as between channels. At high frequencies (such as the 183.0 GHz channels) SSMIS will generally see bigger scattering depressions than GMI due to the different polarizations; if this effect is not represented in the observation

operator, it degrades the consistency of the DA.

## 4  Discussion

At mm/sub-mm wavelengths, the interaction between ice hydrometeors and radiation is chiefly driven by scattering (high single scattering albedo, see, e.g., Eriksson et al., 2011). Consequently, such interaction induces a considerable polarization signature that strongly depends on the size, shape, and orientation of the hydrometeors. In other words, non-spherical ice hydrometeors

are characterized by non-unit aspect ratios, i.e., the ratio of the longest to the shortest axis, and, if they are large enough, they tend to be oriented in the atmosphere under relatively low turbulence conditions (e.g., stratiform regime or anvil regions of convections). This results in viewing-dependent scattering properties, leading to brightness temperature differences between V- and H-polarized channels, i.e., the polarization difference.

### 4.1  Polarization differences due to oriented ice hydrometeors

In the GMI observations examined here, the high frequency dual-polarization channels (89.0 GHz and, particularly, 166.5 GHz) show a global arch-like relationship between PD and $T_{\mathrm{BV}}$, confirming the findings of Gong and Wu (2017) and Brath et al.

(2020) over the same frequencies and Defer et al. (2014) at the slightly lower frequency of MADRAS, i.e., 157.0 GHz. The arch-like relationship is generally attributed to two processes: the saturation of polarisation under conditions of multiple scattering and the composition of the hydrometeors (size, shape, and orientation) within a dynamic environment. Increasing the number of large enough ($\approx$ 40–150 $\mu$m depending on shape) horizontally oriented non-spherical ice hydrometeors (e.g., flat plates, columns, and fluffy snow aggregates) under relatively low turbulence conditions (e.g., stratiform regime or anvil regions of convections) leads to an increasing scattering, and hence, a stronger polarization signal (e.g., Spencer et al., 1989; Gong and Wu, 2017; Gong et al., 2020; Brath et al., 2020). However, the final polarization state results from only the first few orders of scattering (similar effects are seen at visible frequencies, see, e.g., Barlakas (2016) and references therein). Hence, an increasing multiple scattering process due to the presence of enough hydrometeors will lead, at first, to a saturation (plateau) of the polarization and, at second, to a further decrease of the PD. Accordingly, the low PD values are found at warm $T_{\mathrm{BV}}$ ($\approx$ 275 K), corresponding to very thin clouds, while the largest values are linked to intermediate cold $T_{\mathrm{BV}}$ and medium thick clouds, particularly in the anvil regions of convection. Within deep convective cores (at increasingly low $T_{\mathrm{BV}}$), tumbling motions may lead to the formation of less oblate hydrometeors (i.e., hail and graupel, Jung et al., 2008), disrupt any hydrometeor orientation inducing either higher tilt angles (see Fig. 1) or even total random orientation, and together with the enhanced multiple-scattering process, lead to low or absent PD ($\approx 0 - 7$ K) (e.g., Spencer et al., 1989; Gong and Wu, 2017; Gong et al., 2020).

In contrast to 89.0 GHz (see Fig. A1a), the arch-like shape of PD $- T_{\mathrm{BV}}$ at 166.5 GHz (see Fig. 4a) is characterized by a greater dynamical range ($\approx$140–275 K compared to $\approx$170–275 K), in line with the results reported by Gong and Wu (2017), Galligani et al. (2021), and Defer et al. (2014), and peaks at larger PD values ($\approx 14$ K compared to $\approx 10$ K). The lower PD found at 89.0 GHz could be explained by its multi-sensitivity aspect; a frequency of 89 GHz is quite sensitive to water vapor, water cloud, and rain droplets that could diminish the polarization signal due to emission, especially if there is a liquid layer below the ice hydrometeor layer.

## 4.2 Approximate treatment of polarized scattering from oriented ice hydrometeors

In the current framework of RTTOV-SCATT, only TRO hydrometeors are considered, which fails to reproduce the observed PDs, leading to errors in polarized scattering up to about 10–15 K. To model the effect of preferentially oriented (ARO) ice hydrometeors, a simple correction scheme has been implemented in RTTOV-SCATT that reduces TRO extinction in V-polarized channels and increases it in H-polarized ones. This is quantified by the ratio of extinction in H- over that in V-channels, i.e., the polarization ratio $\rho$. This ratio holds an indirect microphysical representation of the aspect ratio of non-spherical oriented hydrometeors.

Totally randomly oriented hydrometeors (control run) are characterized by a $\rho$ of 1.0 and cannot induce polarization in our modelling framework. For the dual polarization channels, the value of $\rho$ that approximates best the orientation of non-spherical ice hydrometeors and yields a more symmetrical error is 1.5 and 1.4 for 89.0 and 166.5 GHz, respectively. For example, the mean error in the differences in PD between simulations and observations ($\mathrm{PD^o - PD^b}$) is diminished by an order of magnitude at 166.5 GHz. These values of $\rho$ are equivalent to adjusting the extinction of the control run by $\pm 20\%$ ($\alpha = 0.20$) at 89.0 GHz

and by $\pm 16.7\,\%$ ($\alpha = 0.167$) at $166.5\,\mathrm{GHz}$ in order to match the extinction at the two polarization components. In addition, at $166.5\,\mathrm{GHz}$, it approximates quite well the magnitude of the $K_{12}$ element of the extinction matrix in case of ARO ($K_{\mathrm{ARO,12}}$) at a tilt angle of $0\,^\circ$ (see Fig 1b; in case of large plate aggregates), which describes the actual differences in the extinction between the H- and V-polarization. Recall here that $\alpha$ in Eq. (6) approximates the magnitude of $K_{\mathrm{ARO,12}}$ at earth incident angles around
$55\,^\circ$. These findings are in line with those reported by Brath et al. (2020). The values of $\rho$ are valid at a global scale (over both land and ocean), with a higher confidence at $166.5\,\mathrm{GHz}$.

Figure 4b shows that, for a $\rho$ value of 1.4, the polarized correction scheme in RTTOV-SCATT is able to do a reasonable job of reproducing the observed scattering arches, but compared to observations it tends to over-favour larger PDs at the peak of the arch (e.g., at $200\,\mathrm{K}$) or low PD values (0–2 K) at the lower part of the arch (e.g., $225 < T_{\mathrm{BV}} < 250\,\mathrm{K}$), while PDs do
not drop low enough at lower $T_{\mathrm{BV}}$ (e.g., $150\,\mathrm{K}$). RTTOV-SCATT does not simulate the full arch-like relationship because it cannot transfer energy from one polarization to the other (the multiple scattering effect). In addition, Geer (2021) reported that the combination of the IFS and RTTOV-SCATT does not simulate deep enough brightness temperature depressions in tropical convection over land (see also Fig. 5a), likely due to insufficient horizontal spreading of the upper glaciated parts of the convective cloud; these scenes, if represented correctly, should have lower PDs according to the hypothesis that turbulence
in the deep convective core is responsible for random orientation and hence depolarisation. However, it does reproduce some of the drop in polarization in strongly scattering scenes which is likely due to saturation of the scattering; the differences between $\tau_{\mathrm{H}}$ and $\tau_{\mathrm{V}}$ become irrelevant. Ideally, the choice of $\rho$ would be situation dependent. However, this would increase the intricacy of the forward operator and further complicates any attempts to impartially certify the impact of such a correction scheme.

Going back to the optimum choice of the $\rho$, at $89.0\,\mathrm{GHz}$, a more symmetrical error is achieved at a ratio of 1.45 over the
ocean, but 1.5 overall, but this could be attributed to the relatively low number of pixels fulfilling the rather strict screening method which minimizes the strongly polarizing ocean surface. The IFS simulates (with slightly low confidence) this channel quite differently over land and ocean. Partly, this is due to its aforementioned multi-sensitivity aspect that complicates radiative transfer. Accordingly, some polarization signal simulated at this frequency could originate from strongly polarizing inland water, e.g., large lakes or flooding, that have not been perfectly screened, or even due to shallow clouds over ocean at high
latitudes. Similar patterns of potential surface contamination have been recently reported by Galligani et al. (2021). Furthermore, liquid droplets can be horizontally aligned inducing small PDs ($\approx 0.5\,\mathrm{K}$, Ekelund et al., 2020) that are observed but not simulated in the IFS. All these, in addition to the known limitations of the IFS and RTTOV-SCATT in representing convective systems (Geer, 2021) could potentially explain the rather large polarization ratio required to obtain reasonable simulations (good fit to the observations) at 89.0GHz.

Given that larger PDs are found at $166.5\,\mathrm{GHz}$ than at $89.0\,\mathrm{GHz}$, it might seem obvious that there should also be a larger polarisation ratio at $166.5\,\mathrm{GHz}$. But this makes the incorrect assumptions that the level of extinction is the same for both frequencies, and that the frequencies are sensitive to the same size range of hydrometeors. In fact, the PD has a complex dependence on parameters such as size, shape, aspect ratio, PSD, along with the channel's frequency and the level of extinction (Xie and Miao, 2011; Defer et al., 2014). Lower frequencies are more sensitive to larger hydrometeors (e.g., Buehler et al.,
2007) and it is likely that the axial ratio and thus the orientation increases with size. This effect would suggest a larger PD at

89.0 GHz. Further, the extinction generated by ice hydrometeors is smaller at 89.0 GHz, so if all else were constant, to obtain the same PD at 89.0 GHz as at 166.5 GHz, it would also require a higher polarisation ratio at 89.0 GHz. Putting aside the small differences between the best polarisation ratio at each frequency, the values found in this study are reasonably consistent with other studies, such as the ratios of 1.2–1.4 that are reported by Gong and Wu (2017) for the same dual polarization channels of GMI. In the studies of Davis et al. (2005) and Defer et al. (2014), it is not the polarisation ratio but the actual microphysical aspect ratio that is reported, so it is hard to compare exactly. However, Davis et al. (2005) found similar aspect ratios at the lower frequency channel of 122.0 GHz of the Earth Observing System (EOS) Microwave Limb Sounder (MLS). On the other hand, Defer et al. (2014) suggested an aspect ratio of 1.6 to be the most realistic one in reproducing the PDs observed by MADRAS in both 89.0 and 157.0 GHz. In any case, these ratios are all subject to the microphysical representation of the hydrometeors employed, so a hydrometeor type with a different scattering efficiency would result in a different ratio.

## 4.3   Linking the polarization difference to a hydrometeor type

The polarization signal prompted by preferentially oriented hydrometeors strongly depends on their microphysical representation. To further interpret this signal, $\rho$ was applied to each ice hydrometeor individually. Horizontally oriented large scale snow and deep convective snow (graupel) are responsible for causing most of the observed PDs at 166.5 GHz, in consistency with Brath et al. (2020), with snow being associated with the largest PD values as also addressed by Defer et al. (2014) and Gong and Wu (2017). Comparing the snow- and graupel-only scaled simulations, the former one produces PD values across the entire range, while the latter one is responsible for generating mostly low to medium PDs at warm to intermediate cold $T_{\mathrm{BV}}$ (270 to 210 K). This is thought to come from the representation of convection in the forecast model and RTTOV-SCATT as occupying only 5% of the model grid-box. Even if a convective column generates strongly polarized scattering in RTTOV-SCATT, this limits its effect on the whole-scene brightness temperatures. In contrast, precipitation generated by the large-scale scheme typically occupies a large fraction of the grid-box and is able to generate stronger PDs in the complete scene.

In reality, similar behaviors are observed, but driven by different processes (Gong and Wu, 2017; Gong et al., 2020). In case of snow, the tumbling motions in the ambient environment and the two-fold scattering effect described in Sect. 4.1 can explain the observed polarization patterns. A plausible interpretation of the different polarization signal induced by graupel is its higher weight and the tumbling motions within deep convective cores that lead to less oblate shapes (Jung et al., 2008); for the same amount of hydrometeors it will be less oriented, resulting in lower PD values.

Although one could assume that cloud ice hydrometeors are too small to induce any polarization at 166.5 GHz, the cloud ice only scaled simulations conducted here suggest a minor PD (less than 1.2 K) with a preference to negative values at low brightness temperatures. Representing the cloud ice by the large column aggregate habit and the PSD introduced by Heymsfield et al. (2013) can potentially lead to large enough hydrometeors, and hence, to enough scattering to induce a visible polarization signal. This supports the suggestion by Gong et al. (2020) that some smaller PDs at 166.5 GHz may come from cloud ice. At warmer temperatures, negative PDs at 166.5 GHz were reported by Gong and Wu (2017); they linked this to clear sky conditions and instrumental noise. At colder temperatures, they also saw negative PDs, without providing any clear explanation. Here, negative PD values are simulated in deep convective areas ($T_{\mathrm{BV}}$ below 180 K). The most likely explanation

is that RTTOV-SCATT is representing cloud ice above deep convection with relatively low single-scattering albedos, and hence, there is an emission signal that is stronger in the H-channels due to the stronger extinction. In reality, negative PDs are also measured (Davis et al., 2005; Prigent et al., 2005), with the most dominant interpretation being the vertical orientation of hydrometeors. A likely explanation of the vertical orientation is lightning activities at cloud top (hydrometeor electrification) of deep convective systems (Prigent et al., 2005). However, RTTOV-SCATT does not represent vertically oriented hydrometeors,

so if it is able to simulate a negative PD through absorption, this suggests the electrification hypothesis may only be one part of the story.

## 5    Conclusions

Herein, an effort has been carried out to improve the physical representation of polarized scattering in RTTOV-SCATT (Radiative Transfer model for TOVS that accounts multiple scattering) and to explore whether such an improvement would have an

impact on the forecast of the European Centre for Medium-Range Weather Forecasts (ECMWF). To approximate the effect of oriented ice hydrometeors in reproducing the observed brightness temperature ($T_B$) differences between vertical (V) and horizontal (H) polarization, i.e., the polarization difference (PD $= T_{BV} - T_{BH}$), from conical scanning sounders, the layer optical thickness of these hydrometeors is increased in H- and decreased in V-channels. This is governed by the assumed polarization ratio ($\rho$).

By optimising measures of fit between dual-polarization observations from the Global Precipitation Mission (GPM) microwave imager (GMI) and simulations from the Integrated Forecast System (IFS) of ECMWF, it follows that the value of $\rho$ that models best the orientation of ice hydrometeors is 1.5 at 89.0 GHz and 1.4 at 166.5 GHz, with lower confidence at 89.0 GHz. With these settings, RTTOV-SCATT is capable of simulating the effect of oriented hydrometeors and it generates a reasonable representation of the observed arch-like relationship between PD and $T_{BV}$ (or $T_{BH}$). This reduces maximum mod-

elling errors in PD by about 10–15 K. Although the simulated PD is not perfect, the discrepancies in PD between observations and simulations are ameliorated, by an order of magnitude at 166.5 GHz, and the remaining errors are now approximately symmetrical. In the context of data assimilation (DA), assigned observation errors are quite large (e.g., up to 35 K in terms of the PD for the GMI 166.5 GHz channels in deep convective areas), but a 15 K error reduction would still be a significant improvement.

Applying $\rho$ to each ice hydrometeor type individually, we demonstrated that the polarization signal strongly depends on their microphysical representation. Assuming that IFS gives a fair representation of the real atmosphere, it is suggested that snow and graupel are responsible for causing most of the observed polarization signal, with snow producing PD values across the entire range and graupel generating mostly low to medium PDs at warm to intermediate cold brightness temperatures. Where only cloud ice was polarized, simulations give a negative polarization signal over deep convection or heavy precipitation, that

could be potentially linked to enhanced emission effects in the H-channels.

Cycling DA experiments were used to examine the impact of representing polarized scattering in the observation operators for the conical scanners: GMI, Special Sensor Microwave Imager/Sounder (SSMIS), and Advanced Microwave Scanning

Radiometer-2 (AMSR-2). This was conducted for a polarization ratio of 1.4 applied to all ice species and all channels. Validation against independent references (i.e., instruments employed in DA, but where the observation operator was unchanged; clear-sky DA) showed mixed but broadly neutral changes to forecast quality – in other words, the improved modelling of polarization difference does not appear to affect the broader forecast quality. However, some semi-independent validation was possible using the conical scanners to which the polarization correction was applied: this showed that when the desired $\rho$ was achieved with a unilateral scaling, e.g., in V-channels, it was possible to see positive impacts in the forecast skill measured by the conical sounder channels in H-channels, and vice versa. This suggests that representing polarization makes the forward operator more consistent between V- and H-channels of the same instrument, but also between instruments with different polarizations. For example at high frequencies (e.g., $\approx 183.0$ GHz), SSMIS will generally see bigger scattering depressions than GMI due to the different polarizations. So, if SSMIS and GMI both observe the same feature (even at different times during the assimilation window, or in different windows), then the representation of polarized scattering would allow the DA scheme to make more consistent corrections to the forecast. This likely improves the consistency of the forecast in areas of cloud and precipitation (e.g., frontal areas).

A second implication from the cycling DA experiments came from noting that a unilateral decrease of scattering in V-channels gave marginally better results (versus independent humidity-sensitive observations), whereas the unilateral increase in H-channels gave marginally worse results. This suggests the need for tuning the overall level of scattering in RTTOV-SCATT. This is particularly necessary since the selection of hydrometeor assumptions used for this work was tuned against instruments with predominantly H-polarization channels at high frequencies (e.g., Geer, 2021, used SSMIS). Hence, Geer (2021) carried out a further retuning of the microphysical assumptions, taking account of the polarization scheme introduced here, with a default $\rho$ value of 1.4 for all ice species at all channels. That final retuning, along with the new polarization scheme, provides the final configuration for RTTOV-SCATT version 13 (v13.0). This configuration is also aimed at implementation in a future cycle of the IFS.

The performance of this polarization scheme has been tested only with conical scanning sounders and it is likely valid only at earth incident angles around $55^{\circ}$, where hydrometeor orientation does not change the overall level of extinction (the $K_{11}$ term in the Brath et al. (2020) database, as presented in Sec. 2.5). To model the effect of hydrometeor orientation at other angles will require significant further work, since both the overall extinction and the degree of polarization must vary as a function of the earth incident angle. For cross-track microwave sounders this will be particularly difficult, since they are not currently equipped with high-frequency dual-polarization channels with which to validate the results, and the polarization rotates across the swath. Future work could also aim at developing a similar correction scheme for radar backscattering. A subsequent step would be to explore the performance of $\rho$ at higher sub-millimeter wavelengths; the upcoming Ice Cloud Imager (ICI) mission, with dual-polarization channels at frequencies that are sensitive to the column scattering due to ice hydrometeors (243.2 and 664.4 GHz), will provide further insights regarding the polarization signal due to small oriented ice hydrometeors.

*Code and data availability.* The web address for RTTOV-SCATT is nwp-saf.eumetsat.int/site/software/rttov, where the model can be obtained through registration. Due to the large volume of data generated by the assimilation experiments of the IFS, its permanent archiving/curation is not possible. But, reanalysis data, e.g., ERA-5 (Fifth generation of ECMWF atmospheric reanalyses of the global climate), is available.

## Appendix A: Screening method

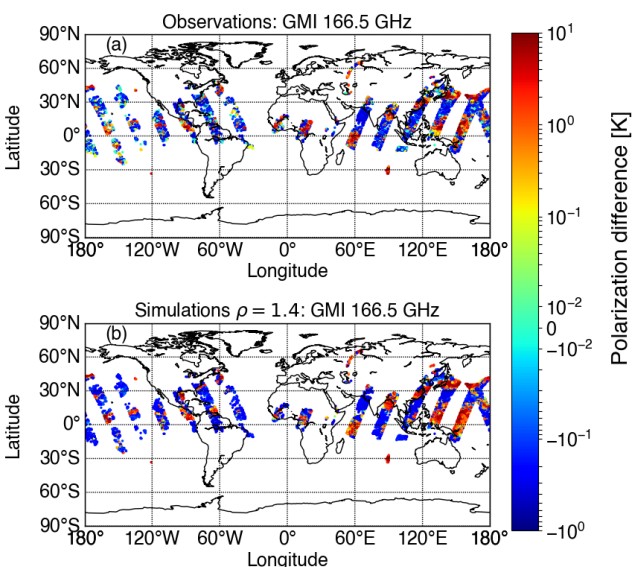

**Figure A1.** Performance of the screening method at an example scene measured by GMI on 13 July 2019 at a frequency of 166.5 GHz. Polarization difference, i.e., brightness temperature differences between the vertical and horizontal polarization ($\text{PD} = T_{\text{BV}} - T_{\text{BH}}$), due to oriented ice hydrometeors as (a) observed ($\text{PD}^{\text{o}}$) by GMI and (b) simulated ($\text{PD}^{\text{b}}$) for a polarization ratio ($\rho$) of 1.4.

To minimize the surface contribution, the clear-sky surface-to-space transmittance ($t$) simulated by RTTOV has been employed. Recall here that the superscript b corresponds to the simulated (background) quantities. Accordingly, the surface contribution to the polarization can be approximated (ignoring the downward radiance term) by:

$$\text{PD}^{\text{b}}_{\text{sfc}} = T^{\text{b}}_{\text{BV,sfc}} - T^{\text{b}}_{\text{BH,sfc}} \simeq (e_{\text{V}} - e_{\text{H}}) \cdot T_{\text{sfc}} \cdot t_{166.5}. \tag{A1}$$

Where $e_{\text{V}}$ and $e_{\text{H}}$ are the surface emissivities at V- and H-polarization, respectively, $T_{\text{sfc}}$ is the surface temperature, and $t_{166.5}$
denotes the transmittance at 166.5 GHz. An upper threshold $t_{166.5}$ value of 0.05 has been adopted to mask out the surface contribution at both dual-polarization channels. This can be translated (via Eq. A1) to a maximum surface induced polarization difference of 2.9 K. Lower values of the clear-sky $t_{166.5}$ have been also tested, i.e., 0.01–0.04 (0.58 K–2.32 K in $\text{PD}^{\text{b}}_{\text{sfc}}$), to ensure that the polarization signature originates from oriented ice hydrometeors and not from clear-sky conditions and thus, from the surface. To isolate cloudy GMI pixels, we additionally defined the hydrometeor impact as:

$$\Delta T^{\text{o}}_{\text{B}} = T^{\text{o}}_{\text{B,cloudy}} - T^{\text{b}}_{\text{B,clear}},$$
$$\Delta T^{\text{b}}_{\text{B}} = T^{\text{b}}_{\text{B,cloudy}} - T^{\text{b}}_{\text{B,clear}}. \tag{A2}$$

$\Delta T^{\text{o}}_{\text{B}}$ ($\Delta T^{\text{b}}_{\text{B}}$) represents the hydrometeor impact defined by the observed (simulated) brightness temperature in all-sky conditions minus the simulated equivalent atmosphere without clouds. Any situation with a hydrometeor impact above 0 K is

rejected; this is done on a channel-wise basis and the rejection is done if either observations or simulations cross this threshold. The bias correction is included at 166.5 GHz, but excluded at 89.0 GHz because it led to a mismatch between observations and simulations. This two-fold screening method does not completely eliminate surface polarization signatures at 89.0 GHz. Hence, an additional basic filtering has been applied to this channel only. First, pixels where the control run yields a $PD^b$ greater than 1 K are masked out, since the control run does not produce any polarization due to preferentially oriented hydrometeors. Second, pixels where the observed PD ($PD^o$) is greater than 5 K and $T^o_{BV} > 265$ K are excluded (this is the visible rectangular bite out of the distribution in Fig. B1a).

Figure A1a displays the performance of the screening method at a scene measured by GMI on 13 July 2019 at 166.5 GHz. This method minimises any surface contamination, albeit it is rather strict and screens out thin cirrus clouds, especially over the mid-latitudes. Figure A1b demonstrates the rather good performance of the polarization ratio ($\rho$) of 1.4 to simulate $PD^o$.

## Appendix B: Two dimensional histograms of the arch-like relationship at 89.0 GHz

Figure B1 displays the two dimensional (2D) histogram of the arch-like relationship between the polarization difference and the brightness temperature at V-polarization at 89.0 GHz as observed by GMI (Fig. B1a), simulated for a $\rho$ of 1.5 (Fig. B1b), and the corresponding 2D histogram divergence (Fig. B1c). Simulations fail to reproduce the full arch-like shape. Due to the screening method, most moderately cloudy scenes over the ocean have been excluded, leading to a rather small sample; but, the main $PD^o$ branch ($230 < T_{BV} < 275$ K) and the $PD^o$ peak ($\approx 10$ K) have been modeled quite well by a $\rho$ value of 1.5.

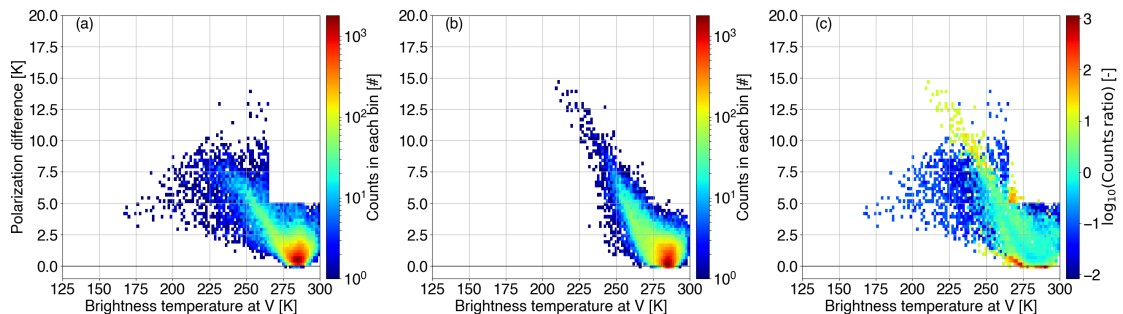

**Figure B1.** Two dimensional (2D) histograms describing the arch-like relationship between the polarization difference and the brightness temperature at V-polarization at 89.0 GHz as (a) observed ($PD^o - T^o_{BV}$) by GMI, (b) simulated ($PD^b - T^b_{BV}$) for a polarization ratio of 1.5, and (c) the 2D histogram divergence between ($PD^o - T^o_{BV}$) and ($PD^b - T^b_{BV}$). In panel (c), white areas denote the case where both the observed and the simulated 2D bins are empty.

*Author contributions.* The work was conceptualised and refined by VB, AJG, and PE. Writing, coding, and analysis was mainly carried out by VB, with contributions from AJG.

*Competing interests.* The authors declare no conflict of interest.

*Acknowledgements.* The work of Vasileios Barlakas at Chalmers University of Technology is funded by a EUMETSAT fellowship program. Niels Bormann, Luisa Ickes, Franz Kanngiesser, Inderpreet Kaur, and Simon Pfreundschuh are thanked for the many thoughtful comments 670 that led to the overall improvement of the manuscript.

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
