# Peer review of "Introducing hydrometeor orientation into all-sky microwave/sub-millimeter assimilation"

_Atmospheric Measurement Techniques, 2020_

## Referee Comment (RC1) · Anonymous Referee #1 · 22 Dec 2020

Overview The manuscript "Introducing hydrometeor orientation into all-sky microwave/sub-millimeter assimilation" describes an approach to improve agreement between simulated and observed polarized microwave brightness temperatures in the assimilation of cloudy remote sensing data. In lieu of implementing a fully-polarized radiative transfer model into the data assimilation scheme, the authors propose to adjust the extinction of "unpolarized" cloud and precipitation scattering properties with an extinction ratio. Such a characterization has sufficient rooting in the body of literature, and it is based on the knowledge that first order scattering effects are the primary mechanism by which clouds and precipitation modify propagating electromagnetic waves. While the overarching approach is well-grounded, there are significant gaps that need to be addressed. Significant information needed to assess

the manuscript is missing. The implementation itself is presented more as a fitting exercise with very loose connections to cloud and precipitation physics and the related radiative transfer. The parameterization of the extinction ratio is a bit simplistic, and its application is overgeneralized.

Specific comments Lines 92-98: The "microphysical setup" that is used for the RTTOV forward operator is based on manuscript that is in preparation (Geer 2020), and not available to consider for the review. Salient details are missing, and those details are necessary to properly review this manuscript. Some of this information is basic: What are the size ranges for the different ice habits? But, there are deeper details that need to be considered. The use of a model like Liu's sector snowflake to cover a broad range of ice types makes sense. However, expanding the ice microphysics to multiple ice habits, each with associated high-fidelity scattering properties, offers an opportunity implement a microphysical approach that has a physical basis. While these ice models may be the best numerical matches, the morphologies of the habits selected don't intuitively map to the stated ice classes, with the exception of cloud ice, and there is very little insight into the selection processes.

Table 1: The reference for the sector snowflake should be Liu 2008. Also, following the previous comment, please add the size ranges for each liquid and ice habit.

Lines 154-155: The process by which hydrometeors align, and how size relates to alignment needs to be considered in significantly more detail here.

Line 159-161: The salient details of [Brath et al., 2020] that are applicable to this manuscript, i.e., a description of the geometry including particle and laboratory reference frames, need to be included. There needs to be enough information to make this paper understandable on its own. As part of this, "tilt" needs to be defined.

Line 203: One month of data doesn't seem sufficient to represent the full range of brightness temperatures and polarization differences.

Line 235: Based solely on Figure 2, why not 1.2? This is certainly within the distribution of the observations, and follows the trend of the observations. especially for combinations of low Tbv and low PD.

Line 259: The higher ratio for 89 GHz compared to 166 GHz doesn't make physical sense, given that polarization differences at 166 GHz are noticeably higher (and the plot in the appendix suggests that the agreement degrades with increased ice scattering).

Figure 4: Looking at this plot, the distribution seems a bit bifurcated (but this *could* be an artifact of insufficient data, referencing previous comment for line 203): one high-slope relationship that contains a large bulk of the data, and another that exhibits significantly more downward curvature with increasing ice scattering. It's also the curve with the lowest Tbv values. The analysis presented in this paper seems to be fitting to the high slope data. It would be instructive to also plot something like 5b here.

Figure 5: This plot doesn't offer much since this is really a bi-variate distribution, or even better, it should be paired with the mono-variate histograms of Tbv.

Section 3.4: Overall, this section is a bit confusing, and it doesn't add much in it's current form. Referring to the radiometer bands as numbers (instead of frequency and polarization) definitely make it more opaque. Also, why ATMS? The consideration of polarization is significantly more complicated than the conical scanner, and the polarization ratio has not been characterized over a range of incidence angles which means that the method developed in this paper is not applicable to most of the data.

Line 334: reducing extinction, not scattering

Lines 340 and 341: again extinction, not scattering

Section 4.1, first paragraph: The description of why the polarization differences approach zero again for deep convection is incomplete. Yes, multiple scattering depolarizes the radiation; however, particle morphologies and orientations within a dynamic environment are also at play: spherical (or less oblate) hail and graupel that may be

tumbling in the turbulent environments in which they form, although these processes are not well understood [Jung et al 2009].

References Jung, Y., Zhang, G., & Xue, M. (2008). Assimilation of Simulated Polarimetric Radar Data for a Convective Storm Using the Ensemble Kalman Filter. Part I: Observation Operators for Reflectivity and Polarimetric Variables, Monthly Weather Review, 136(6), 2228-2245.

---

## Referee Comment (RC2) · Anonymous Referee #2 · 25 Dec 2020

Review comments for "introducing hydrometeor orientation into all-sky microwave/sub-millimeter assimilation" by Barlakas et al.

This manuscript thoroughly studies the feasibility of using a simple and fast strategy to incorporate the frozen hydrometeor orientation induced radiance difference between V- and H- polarized (V-pol and H-pol) channels at high frequency microwave/sub-millimeter (MW/sub-mm) band into the current ECMWF data assimilation (DA) system. This simple strategy involves a modification of the optical depths calculated from model simulated hydrometers at V-pol and H-pol so to make their ratio satisfying a fixed value. This approach is proved in the manuscript to be able to mimic the observed arch-shape of PD-TBv relationship at 166 GHz, and greatly reduce the skewness of O-B distribution so to make the all-sky DA really possible. At 89 GHz, it's found more difficult to

achieve a best-fit because it's also impacted by surface and water emission from below the frozen hydrometer layer. Then, three sets of modifications are tested (modifying V-pol only, V-pol and H-pol together, and H-pol only) for the impact on model forecast. Although the impact is in general neutral, this method allows DA schemes to be more consistent among channels with V-pol and H-pol, as well as across different satellite instruments that have different polarizations at the same channel frequencies.

I enjoyed reading this manuscript. This work is novel and ground-breaking as it's the first research I've ever seen to put effort to DA the frozen hydrometer orientation information. Although currently no GCM microphysics schemes really simulate the orientation characteristics of hydrometers, polarization difference (PD) at high-frequency MW/sub-mm channels is a major hurdle for all-sky DA for these channels mainly because the skewness induced by the microphysics. This work proposes an easy and physically meaningful and consistent way to tackle this problem. The results are very encouraging and meaningful for the preparation of new missions such like ICI as well as opening up a new door to reprocess existing satellite observations. I fully support the publication of this work and looking forward to reading the companion paper (Geer, 2020, to be submitted to AMT).

Still, there are some minor issues that I think require further clarifications. (1) 89 GHz PD, as also discussed in the manuscript, is complicated by not only the surface PD signal contamination, but more importantly, but liquid emission. It is a damping effect if liquid emission from water cloud or rain layer beneath the frozen hydrometer layer is completely random oriented, however, rain droplet tends to be horizontally aligned as well. This adding an extra dimension of difficulty which was not mentioned in the paper, and not considered at least in full RTM simulations. I would use a lot of caution of applying a best-fit ratio to 89 GHz.

(2) The best-fit ratio is achieved globally on a statistical sense, and a fixed value is applied globally. In reality, I would imagine it should vary by weather systems and/or locations. For example, snow crystal shape, size and orientation would be different behind the cold front versus ahead of it; analogously, snow characteristics in an Arctic low should be different from those in a tropical deep convective system. Can the value of rho be latitudinal varying or weather regime dependent (e.g., convective versus stratiform pixels in GCM grid). I'm not asking to perform these analysis, but I'd like to see authors' response on this question: in other words, would a varying rho be potentially more beneficial to the DA from the satellite retrieval perspective?

(3) Other than the impact on forecast, what are the impact on other variables, for example, total column IWP (all frozen hydrometers), TOA radiation budget, etc.? I doubt whether a discernable impact but it would be nice if these "climate" impacts could be discussed or at least mentioned. In the future, if model physics start to include orientation impact on, e.g., radiation, or depositional growth of particles, I would imagine water cycle and the radiation budget would be impacted eventually.

Minor comments: L154: "if they are large enough, they tend to be oriented". This is not quite correct. Only if the aspect ratios are large (i.e., flatter) and the ambient environment flow is relatively stable (e.g., stratiform regime), that large frozen hydrometers tend to be oriented in a predominant direction. In some cases DPR's DWR indicated big-sized particles but collocated GMI 166 GHz PD signals are small.

Figure 3, top right panel: it looks divergence trend hasn't reached a minimum by rho=1.5 yet. Also, for these statistics, are surface-contaminated pixels removed? If yes, I'm a big confused why ocean and land skewnesses are so distinctly different at 89 GHz.

L440: just a comment – I like your discussions here. Several possibilities are presented, and you leave some room for future exploration. Actually, we've tried to connect collocated lightning data we GMI negative PD signals but failed to establish a statistically robust relationship. Maybe it's simply because lightning happens at instantaneous time-scale that typical collocation criteria (10-15 mins time difference) doesn't work, but geographical distribution of negative PD also doesn't direct point to an as-

sociation with lightning. I honestly doubt in real world, cloud ice could generate a cold 166 GHz TB as cold as 125 K (e.g., your Fig. 6c), which means tremendous number density and extremely large plate-type of cloud ice. As CALIPSO only sees 1-5% of chances of horizontally oriented ice globally, I believe cloud ice orientation doesn't happen as often as snow aggregates, and it's impact should be minimal at 166 GHz.

---

## Author Comment (AC1) · 11 Feb 2021

**Response to Referee #1**

To begin with, we would like to thank the anonymous referee #1 for her/his time and efforts in reviewing our submitted manuscript. We also thank the anonymous referee #1 for her/his constructive comments and suggestions that certainly will improve the manuscript significantly. Finally, we would like to inform the anonymous referee #1 that we will revise the manuscript according to her/his comments and the comments of the anonymous referee #2. Below we respond to the main questions/comments raised by the referee, and outline how we will revise the manuscript. To that end:

- referee's comments are given in blue,

- our responses are outlined in normal format, and

- **any suggested textual changes are given in bold format**.

**Responses to general comments (GC) from referee #1 (GC1)**

(**GC1.1**) In lieu of implementing a fully- polarized radiative transfer model into the data assimilation scheme, the authors propose to adjust the extinction of "unpolarized" cloud and precipitation scattering properties with an extinction ratio. Such a characterization has sufficient rooting in the body of literature, and it is based on the knowledge that first order scattering effects are the primary mechanism by which clouds and precipitation modify propagating electromagnetic waves. While the overarching approach is well-grounded, there are significant gaps that need to be addressed. The implementation itself is presented more as a fitting exercise with very loose connections to cloud and precipitation physics and the related radiative transfer. The parameterization of the extinction ratio is a bit simplistic, and its application is overgeneralized.

Let us try to summarize our view: This work comprises a first attempt to introduce hydrometeor orientation within a data assimilation (DA) context by improving the physical representation of polarized scattering that was until now completely ignored. Currently, it is too expensive to move towards a fully-polarized radiative transfer in DA, since scattering due to oriented non-spherical hydrometeors increases the complexity of all-sky simulations and substantially increase the computational demands even at non-operational radiative transfer solvers. We introduce an alternative, but physical-based approach that adjusts the extinction as a function of polarization, driven by a special characteristic of the extinction matrix at earth incident angles around 55 ° (zenithal angle of conical scanners): hydrometeor orientation does not change the overall level of extinction. Even if simplified and generalized, this is still a major step forward compared to the situation where the hydrometeor orientation is completely ignored. In the context of fast radiative transfer modelling, particularly in all-sky DA, a special emphasis is given on improving the best fit between model and observations on a global basis and ensuring that improving one aspect does not

lead to a degradation of any other aspect of the DA system (e.g., Geer and Baordo, 2014; Fox, 2020). This context requires
simple solutions that could be applied globally. Ideally, the choice of the microphysical representation of the hydrometeors and
the selection of the polarization factor would be situation dependent. However, this would increase the intricacy of the forward
operator and further complicate any attempts to impartially certify the impact of our choices.

We understand that there are gaps that need to be addressed and hence, we propose to revise the manuscript accordingly,
while providing additional clarifications when needed.

**Responses to specific comments (SC) from referee #1 (SC1)**

(**SC1.1**) Specific comments Lines 92-98: The "microphysical setup" that is used for the RTTOV forward operator is based on
manuscript that is in preparation (Geer, 2021), and not available to consider for the review. Salient details are missing, and
those details are necessary to properly review this manuscript. Some of this information is basic: What are the size ranges
for the different ice habits? But, there are deeper details that need to be considered. The use of a model like Liu's sector
snowflake to cover a broad range of ice types makes sense. However, expanding the ice microphysics to multiple ice habits,
each with associated high-fidelity scattering properties, offers an opportunity implement a microphysical approach that has a
physical basis. While these ice models may be the best numerical matches, the morphologies of the habits selected don't in-
tuitively map to the stated ice classes, with the exception of cloud ice, and there is very little insight into the selection processes.

We feel sorry for any inconvenience caused by the non-availability of Geer (2021), but unforeseen circumstances postponed
its submission. Over the last years, several improvements have been implemented in RTTOV-SCATT, the part of RTTOV that
accounts for multiple scattering. All these led to RTTOV v13 offering a wider and more physical range of ice hydrometeors
(e.g., Eriksson et al., 2018) and particle size distributions (e.g., McFarquhar and Heymsfield, 1997; Petty and Huang, 2011;
Heymsfield et al., 2013). Accordingly, the work of Geer (2021) has a dual objective: First, to explore an observation driven
methodology for developing parameterizations for clouds and precipitation and, second, to improve the physical representa-
tion of ice hydrometeors in RTTOV-SCATT in preparation for the future Ice Cloud Imager (Eriksson et al., 2020) that covers
sub-millimetre wavelengths.

In order to fill in any missing parts, we propose to include a separate section, i.e., "Microphysical configuration" and revise
Table 1 as follows:

[revised manuscript text omitted]

90 **(SC1.2)** Table 1: The reference for the sector snowflake should be Liu (2008). Also, following the previous comment, please add the size ranges for each liquid and ice habit.

The reference for the sector snowflake was correct, please see the details given in SC1.1. We will follow the reviewer's comment and include the size ranges of the hydrometeors. Table 1 is the proposed revised table.

95

**(SC1.3)** Lines 154-155: The process by which hydrometeors align, and how size relates to alignment needs to be considered in significantly more detail here.

The referee is absolutely right; our description was incomplete. We propose the following revision:

100

However, ice hydrometeors are characterized by non-spherical shapes and thus non-unit aspect ratios**. This could potentially lead to preferential orientation driven by gravitational and aerodynamical forces (Khvorostyanov and Curry, 2014) or even by electrification processes (lightning activities at deep convective systems, Prigent et al., 2005). Under turbulence-free conditions, small non-spherical hydrometeors (diameters below $\approx 10\,\mu$m) are totally randomly oriented**
105 **owing to Brownian motion (Klett, 1995); but, if they are large enough, they tend to be horizontally oriented as they fall depending on their shape: this holds true for thick plates with a diameter above $\approx 40\,\mu$m (Klett, 1995), while oblate spheroids and thin plates would adopt horizontal orientation at sizes larger than $\approx 100\,\mu$m (Prigent et al., 2005, and references therein) and $\approx 150\,\mu$m (Noel and Sassen, 2005), respectively. However, turbulent effects can easily disrupt any orientation especially for small hydrometeors or introduce a wobbling motion around the horizontal plane at larger**
110 **sizes (10–30 $\mu$m) (Klett, 1995). In addition, tumbling motions in strong turbulent conditions, e.g., within deep convective cores, induce total random orientation (Spencer et al., 1989).**

**(SC1.4)** Line 159-161: The salient details of Brath et al. (2020) that are applicable to this manuscript, i.e., a description of the geometry including particle and laboratory reference frames, need to be included. There needs to be enough information to
115 make this paper understandable on its own. As part of this, "tilt" needs to be defined.

We propose to revise this section as follows:

[revised manuscript text omitted]

**(SC1.5)** Line 203: One month of data doesn't seem sufficient to represent the full range of brightness temperatures and polarization differences.

155

We acknowledge the referee's concerns with regard to insufficient data. However the literature shows that polarisation signatures are fairly consistent through the seasons and across the globe. Gong and Wu (2017) analyzed six months of GMI data (June, July, and August of 2014 and 2015) at 89.0 and 166.5 GHz and reported that the polarization signatures of ice hydrometeors are fairly consistent across all latitudes, with the only exception being the high latitude belt (70–50 $^\circ$ S). In a

160 recent paper, Galligani et al. (2021) analyzed one year (2015) of dual polarization observations from GMI and found that the latitudinal and seasonal variations of the polarization patterns of oriented ice hydrometeors are similar and within the standard deviation around the yearly median of the polarization difference. For completeness, we did explore the sample size during the study. However, we based these results on an earlier, slightly different microphysical setup compared to Table 1, so it would be complex to refer to this in the main work. For a polarization ratio $\rho = 1.3125$, we conducted simulations for an additional

165 month, and the resulting statistics were nearly the same for one or two months. To highlight, Table 2 displays the different statistical metrics for the two different months and the corresponding total statistics at 166.5 GHz. Although there are some differences between the time periods, these would not change the conclusions of the paper. Therefore we are confident that the one month sample is fully sufficient.

Following the referee's comment and to avoid any similar conception by any future reader, we propose to include the fol-

170 lowing clarification:

**For the passive monitoring experiments, the period of one month was chosen on the basis that polarisation signatures of oriented ice hydrometeors are fairly consistent through the seasons and across the globe (Gong and Wu, 2017; Galligani et al., 2021).**

**Table 2.** Statistical metrics, i.e., mean, skewness, and one dimensional (1D) divergence describing the differences in polarization differences between observations and simulations for a polarization ratio of 1.3125, i.e., $\mathrm{PD^o} - \mathrm{PD^b}$, at 166.5 GHz. Statistics are presented for two time periods, i.e., 13 June to 13 July 2019 and 14 July to 13 August 2019, and their total, i.e., 13 June to 13 August 2019. $n$ stands for the sample size.

| Period | $n$ [-] | Mean [K] | Skewness [-] | 1D divergence [-] |
| --- | --- | --- | --- | --- |
| 13 June - 13 July 2019 | 314379 | 0.094 | 1.102 | 0.726 |
| 14 July - 13 August 2019 | 348353 | 0.094 | 1.010 | 0.617 |
| 13 June - 13 August 2019 | 662732 | 0.094 | 1.053 | 0.741 |

175

(SC1.6) Line 235: Based solely on Figure 2, why not 1.2? This is certainly within the distribution of the observations, and follows the trend of the observations. especially for combinations of low Tbv and low PD.

Following the comment of the reviewer, we suggest the following revision:

The remaining panels in Fig. 2, i.e., b–f, depict the ability of $\rho$ to provide realistic simulations of this behaviour, and a first glance indicates that a $\rho$ value between **1.2 and 1.4 could do a reasonable job, since its within the distributions of the observations.**

(SC1.7) Line 259: The higher ratio for 89 GHz compared to 166 GHz doesn't make physical sense, given that polarization differences at 166 GHz are noticeably higher (and the plot in the appendix suggests that the agreement degrades with increased ice scattering).

We agree that this may seem unexpected. Given that larger PDs are found at $165.5\,\mathrm{GHz}$ than at $89.0\,\mathrm{GHz}$, it might seem obvious that there should also be a larger polarisation ratio at $165.5\,\mathrm{GHz}$. But this makes the incorrect assumptions that the level of extinction is the same for both frequencies, and that the frequencies are sensitive to the same size range of hydrometeors. But in general, the PD depends on several parameters (size, shape, aspect ratio, PSD) along with the channel's frequency and the level of extinction (Xie and Miao, 2011; Defer et al., 2014). A frequency of $89.0\,\mathrm{GHz}$ samples the PSD at larger sizes compared to $166.5\,\mathrm{GHz}$. Since the axial ratio and thus the amount of preferential orientation increases with size, this could lead to higher PDs at $89.0\,\mathrm{GHz}$ (e.g., Defer et al., 2014). If the extinction was lower at $89.0\,\mathrm{GHz}$, as it would be expected, this would also require higher polarisation ratios to generate the same polarisation difference.

Further, we did mention that the IFS simulates this channel with lower confidence, partly due to its multi-sensitivity aspect that complicates radiative transfer; a frequency of $89\,\mathrm{GHz}$ is quite sensitive to water vapor, water cloud, and rain droplets. Accordingly, even after the carefully surface-screening methods, some of the polarization signal simulated at this frequency could originate from inland water, e.g., large lakes or flooding, that have not been perfectly screened, or even due to shallow clouds over ocean at high latitudes. Similar patterns of potential surface contamination have been recently reported by Galligani et al. (2021). All these, in addition to the known limitations of the IFS and RTTOV-SCATT in representing convective systems (Geer, 2021) could potentially explain the rather large polarization ratio required to obtain reasonable simulations (good fit to the observations) at $89.0\,\mathrm{GHz}$.

To that end, we have lower confidence in the polarization ratio found at $89\,\mathrm{GHz}$. This is the reason why, in the forecast impact and the final configuration of RTTOV-SCATT v13, the polarization ratio found at $166.5\,\mathrm{GHz}$ was adopted.

Nevertheless, following the referee's comment a cautious revision will be conducted in Sections 4.1 and 4.2 to improve clarity.

**(SC1.8)** Figure 4: Looking at this plot, the distribution seems a bit bifurcated (but this \*could\* be an artifact of insufficient data, referencing previous comment for line 203): one high-slope relationship that contains a large bulk of the data, and another that exhibits significantly more downward curvature with increasing ice scattering. It's also the curve with the lowest Tbv values. The analysis presented in this paper seems to be fitting to the high slope data. It would be instructive to also plot something like 5b here.

As described in Section 4.2, our polarized correction scheme does a reasonable job in reproducing the observed scattering arches, but it is not perfect. RTTOV-SCATT does not simulate the full arch-like relationship because it cannot transfer energy from one polarization to the other (the multiple scattering effect). In addition, Geer (2021) reported that the combination of the IFS and RTTOV-SCATT does not simulate deep enough brightness temperature depressions in tropical convection over land, likely due to insufficient horizontal spreading of the upper glaciated parts of the convective cloud; these scenes, if represented correctly, should have lower PDs according to the hypothesis that turbulence in the deep convective core is responsible for random orientation and hence depolarisation. However, it does reproduce some of the drop in polarization in strongly scattering scenes which is likely due to saturation of the scattering; the differences between $\tau_H$ and $\tau_V$ become irrelevant. Ideally, the choice of the polarization ratio would be situation dependent. However, this would increase the intricacy of the forward operator and further complicates any attempts to impartially certify the impact of our correction scheme. That said, we do fit the high slope data where most of the points are bundled. However, we will revise the corresponding section to improve clarity.

Following the reviewer's recommendation, we will also include the two dimensional (2D) divergence and revise the text accordingly. The revised figures are Figs. R1 and R2. Panels c display the corresponding 2D divergence, highlighting the areas where RTTOV-SCATT underperforms. In the revised manuscript, we will also update Fig. 3 to include the total 2D divergence as a function of the polarization ratio; see Fig. R3. Note here that a minor bug has been corrected at the same time, whereby points with PDs above 15 K were not fully included in all the plots that display histograms or in the divergence calculations; this bug was not affecting the mean or the skewness derivation. The statistics of the total 2D divergence in Fig. R3 suggest that slightly lower polarization ratios lead to the best agreement between observations and simulations, i.e., 1.45 at 89.0 GHz and 1.35 at 166.5 GHz. However, we stick to the original selection of the "best" polarization ratio due to the skewness. Data assimilation assumes that errors are Gaussian and unbiased; hence we prioritize minimising the measure of skewness, rather than the 2D divergence. In addition, the median and 1D divergence are consistent with the skewness. In the revised manuscript, this additional information will be included.

[Figure]

**Figure R1.** Two dimensional (2D) histograms describing the arch-like relationship between the polarization difference and the brightness temperature at V-polarization at $166.5\,\text{GHz}$ as (a) observed ($\text{PD}^{\text{o}} - T^{\text{o}}_{\text{BV}}$) by GMI, (b) simulated ($\text{PD}^{\text{b}} - T^{\text{b}}_{\text{BV}}$) for a polarization ratio of 1.4, and (c) the 2D histogram divergence between $\text{PD}^{\text{o}} - T^{\text{o}}_{\text{BV}}$ and $\text{PD}^{\text{b}} - T^{\text{b}}_{\text{BV}}$. In panel (c), white areas denote the case where both the observed and the simulated 2D bins are empty.

[Figure]

**Figure R2.** As in Fig. R1, but for $89\,\text{GHz}$ and a polarization ratio of 1.5.

[Figure]

**Figure R3.** Statistical metrics, i.e., mean, skewness, and one dimensional (1D) divergence describing the differences in polarization differences between observations and simulations, i.e., $\mathrm{PD^o} - \mathrm{PD^b}$ and the two dimensional (2D) divergence between the observed ($\mathrm{PD^o} - T^o_{\mathrm{BV}}$) and simulated ($\mathrm{PD^b} - T^b_{\mathrm{BV}}$) 2D histograms of the arch-like relationship. Results are presented for $89.0\,\mathrm{GHz}$ (top) and $166.5\,\mathrm{GHz}$ (bottom) over land (in brown triangles), ocean (in blue circles), and globally (in black crosses) for a period of one month (13 June to 13 July 2019) as a function of the polarization ratio ($\rho$). In terms of the control run ($\rho = 1$), the corresponding differences are highlighted in green.

**(SC1.9)** Figure 5: This plot doesn't offer much since this is really a bi-variate distribution, or even better, it should be paired with the mono-variate histograms of Tbv.

Following the reviewer's suggestion, Fig. R4 will represent the updated Fig. 5 (including the minor bug correction), wherein panel (a) highlights the histograms of the brightness temperature at 166.5 GHz as observed ($T_{\mathrm{B}}^{\mathrm{o}}$; in red) by GMI and simulated ($T_{\mathrm{B}}^{\mathrm{b}}$) for a polarization ratio of 1.4 (in blue) and 1.0 (in green; control run); solid and dashed lines denote the brightness temperature at vertical and horizontal polarization, respectively, while the shaded area highlights the their difference. Figure R4a also highlights one of the shortcomings described within the manuscript; there is a need to scrutinize the overall level of extinction within the IFS, since it clearly underestimates brightness temperature depressions in tropical convection.

In the revised manuscript, the additional information will be included.

[Figure]

**Figure R4.** (a) Histograms of the brightness temperature at 166.5 GHz as observed ($T_{\mathrm{B}}^{\mathrm{o}}$; in red) by GMI and simulated ($T_{\mathrm{B}}^{\mathrm{b}}$) for a polarization ratio of 1.4 (in blue) and 1.0 (in green; control run); solid and dashed lines denote the brightness temperature at vertical and horizontal polarization, respectively, while the shaded area highlights their difference. (b) Corresponding polarization differences and (c) the divergence between the distributions of $\mathrm{PD}^{\mathrm{o}}$ and $\mathrm{PD}^{\mathrm{b}}$ as a function of the polarization difference.

**(SC1.10)** Section 3.4: Overall, this section is a bit confusing, and it doesn't add much in it's current form. Referring to the radiometer bands as numbers (instead of frequency and polarization) definitely make it more opaque. Also, why ATMS? The consideration of polarization is significantly more complicated than the conical scanner, and the polarization ratio has not been characterized over a range of incidence angles which means that the method developed in this paper is not applicable to most of the data.

This comment is not totally understood. In the data assimilation (DA) context, it is important to test whether by improving one aspect, we are not degrading another. This is carried out by validation against independent references, meaning instruments (e.g., ATMS) that are employed in DA, but where a change has not been applied. We do agree that accounting for

polarization is much more complicated for cross-track sounders, but we only apply the correction scheme to all the conical
scanners assimilated at ECMWF (three out of the six sensors that are employed in the all-sky assimilation) and where our
scheme is applicable. This is already a big step towards a more physical representation of polarized scattering in DA that was
previously completely ignored, reducing maximum modelling errors in PD by about 10–15 K, while the remaining errors are
now approximately symmetrical.

We do though apologize for any inconvenience caused by referring to some radiometer bands as numbers and hence, in the
revised manuscript, this will be resolved. In addition, and we will further clarify the reference to the independent references.

**(SC1.11)** Line 334: reducing extinction, not scattering.

The text will be revised accordingly.

**(SC1.12)** Lines 340 and 341: again extinction, not scattering.

The text will be revised accordingly.

**(SC1.13)** Section 4.1, first paragraph: The description of why the polarization differences approach zero again for deep convection is incomplete. Yes, multiple scattering depolarizes the radiation; however, particle morphologies and orientations within a dynamic environment are also at play: spherical (or less oblate) hail and graupel that may be tumbling in the turbulent environments in which they form, although these processes are not well understood (Jung et al., 2008).

The referee is absolutely right; our description was incomplete. We propose the following revision:

[revised manuscript text omitted]

---

## Author Comment (AC2) · 11 Feb 2021

**Response to Referee #2**

To begin with, we would like to thank the anonymous referee #2 for her/his time and efforts in reviewing our submitted manuscript and providing feedback. We also would like to express our gratitude for her/his kind words about our manuscript. Finally, we would like to inform the anonymous referee #2 that we will revise the manuscript according to her/his comments and the comments of the anonymous referee #1. Below we respond to the main questions/comments raised by the referee, and outline how we will revise the manuscript. To that end:

- referee's comments are given in blue,

- our responses are outlined in normal format, and

- **any suggested textual changes are given in bold format**.

**Responses to general comments (GC) from referee #2 (GC2)**

(**GC2.1**) 89 GHz PD, as also discussed in the manuscript, is complicated by not only the surface PD signal contamination, but more importantly, but liquid emission. It is a damping effect if liquid emission from water cloud or rain layer beneath the frozen hydrometer layer is completely random oriented, however, rain droplet tends to be horizontally aligned as well. This adding an extra dimension of difficulty which was not mentioned in the paper, and not considered at least in full RTM simulations. I would use a lot of caution of applying a best-fit ratio to 89 GHz.

This is true. We thank the reviewer for pointing out both the damping effect due to liquid emission and the horizontal orientation of liquid spheres (e.g., Ekelund et al., 2020). We will include these suggestions in the revised manuscript.

In fact, due to the high complexity of this channel, in the forecast impact assessment and the final configuration of RTTOV-SCATT v13, the polarization ratio found at 166.5 GHz was adopted. In the revised manuscript, we will further clarify this point.

(**GC2.2**) The best-fit ratio is achieved globally on a statistical sense, and a fixed value is applied globally. In reality, I would imagine it should vary by weather systems and/or locations. For example, snow crystal shape, size and orientation would be different be hind the cold front versus ahead of it; analogously, snow characteristics in an Arctic low should be different from those in a tropical deep convective system. Can the value of rho be latitudinal varying or weather regime dependent (e.g., convective versus stratiform pixels in GCM grid). I'm not asking to perform these analysis, but I'd like to see authors' response on this question: in other words, would a varying rho be potentially more beneficial to the DA from the satellite retrieval perspective?

30   We totally agree that hydrometeor orientation and polarisation should be situation dependent. However, this is more applicable to satellite retrievals, if not to DA yet. In DA, a more intricate scheme would require significant extra tuning work and it would be even harder to validate its performance. But, studies that have looked at latitude and seasonal dependence of the polarisation arches have shown a surprising fair consistency (Gong and Wu, 2017; Galligani et al., 2021). This encouraged us to use a single global fit.

35

**(GC2.3)** Other than the impact on forecast, what are the impact on other variables, for example, total column IWP (all ice hydrometers), TOA radiation budget, etc.? I doubt whether a discernible impact but it would be nice if these "climate" impacts could be discussed or at least mentioned. In the future, if model physics start to include orientation impact on, e.g., radiation, or depositional growth of particles, I would imagine water cycle and the radiation budget would be impacted eventually.

40

The impact of the all-sky assimilation on the hydrometeor mixing ratios is short-lived; any changes in cloud properties in the forecast would come from the response of the model moist physics to any changes in the synoptic situation in response to changes in dynamical variables, and not directly from the assimilated information on hydrometeors. Hence, we are not expecting these variables (e.g., forecast IWP) to be affected given the relatively small impact measured on the main forecast
45   variables. Once parameter estimation can be used to update the forecast model moist physics parameterizations to better fit the all-sky observations, then the benefits to cloud variables can be carried into the forecast.

**Responses to minor comments (MC) from referee #2 (MC2)**

50   **(MC2.1)** L154: "if they are large enough, they tend to be oriented". This is not quite correct. Only if the aspect ratios are large (i.e., flatter) and the ambient environment flow is relatively stable (e.g., stratiform regime), that large frozen hydrometers tend to be oriented in a predominant direction. In some cases DPR's DWR indicated big-sized particles but collocated GMI 166 GHz PD signals are small.

55   The referee is absolutely right; our description was incomplete. We propose the following elaboration:

However, ice hydrometeors are characterized by non-spherical shapes and thus non-unit aspect ratios**. This could potentially lead to preferential orientation driven by gravitational and aerodynamical forces (Khvorostyanov and Curry, 2014) or even by electrification processes (lightning activities at deep convective systems, Prigent et al., 2005). Under**
60   **turbulence-free conditions, small non-spherical hydrometeors (diameters below $\approx 10\,\mu$m) are totally randomly oriented owing to Brownian motion (Klett, 1995); but, if they are large enough, they tend to be horizontally oriented as they fall depending on their shape: this holds true for thick plates with a diameter above $\approx 40\,\mu$m (Klett, 1995), while oblate spheroids and thin plates would adopt horizontal orientation at sizes larger than $\approx 100\,\mu$m (Prigent et al., 2005, and**

references therein) and $\approx 150\,\mu$m (Noel and Sassen, 2005), respectively. However, turbulent effects can easily disrupt any orientation especially for small hydrometeors or introduce a wobbling motion around the horizontal plane at larger sizes (10–30 $\mu$m) (Klett, 1995). In addition, tumbling motions in strong turbulent conditions, e.g., within deep convective cores, induce total random orientation (Spencer et al., 1989).

**(MC2.2)** Figure 3, top right panel: it looks divergence trend hasn't reached a minimum by rho=1.5 yet. Also, for these statistics, are surface-contaminated pixels removed? If yes, I'm a big confused why ocean and land skewnesses are so distinctly different at 89 GHz.

First, we should clarify that ocean (land) represents pixels solely over the ocean (land), excluding pixels across the coastline, while global represents the overall number of pixels (including the coastline). In all figures, results are presented after screening out any surface contamination. In case of 89 GHz, the rather strict two-fold screening method resulted in a rather low sample over the ocean. In addition, the IFS simulates this channel with less confidence over the land compared to the ocean. To highlight, Geer (2021) reported that the combination of the IFS and RTTOV-SCATT does not simulate deep enough brightness temperature depressions in tropical convection over land. These are the reasons why one sees such differences in the statistics between land and ocean at 89 GHz. Note here that a minor bug has been corrected at the same time, whereby points with PDs above 15 K were not fully included in Fig. 5 or in the divergence calculations; however this makes no difference to the other statistical metrics (mean and skewness). Accordingly, the divergence has now a clear miminum that occurs at a polarization ratio of 1.45. But, DA assumes that errors are Gaussian and unbiased; hence we prioritize minimising the measure of skewness.

In the revised manuscript, we are going to provide additional clarifications.

**(MC2.3)** L440: just a comment – I like your discussions here. Several possibilities are presented, and you leave some room for future exploration. Actually, we've tried to connect collocated lightning data we GMI negative PD signals but failed to establish a statistically robust relationship. Maybe it's simply because lightning happens at instantaneous time-scale that typical collocation criteria (10-15 mins time difference) doesn't work, but geographical distribution of negative PD also doesn't direct point to an association with lightning. I honestly doubt in real world, cloud ice could generate a cold 166 GHz TB as cold as 125 K (e.g., your Fig. 6c), which means tremendous number density and extremely large plate-type of cloud ice. As CALIPSO only sees 1-5% of chances of horizontally oriented ice globally, I believe cloud ice orientation doesn't happen as often as snow aggregates, and it's impact should be minimal at 166 GHz.

We agree with the reviewer's comments, and look forward to a future improved understanding of particle orientation, and its dependence on habit, cloud type, and cloud processes.

Best regards,
Vasileios Barlakas, Alan J. Geer, and Patrick Eriksson

**References**

100  Ekelund, R., Eriksson, P., and Pfreundschuh, S.: Using passive and active observations at microwave and sub-millimetre wavelengths to constrain ice particle models, Atmos. Meas. Tech., 13, 501–520, https://doi.org/10.5194/amt-13-501-2020, https://www.atmos-meas-tech.net/13/501/2020/, 2020.

Galligani, V. S., Wang, D., Corrales, P. B., and Prigent, C.: A parameterization of the cloud scattering polarization signal derived from GPM observations for microwave fast radiative transfer models, IEEE Trans. Geosci. Remote Sens., pp. 1–10,

105  https://doi.org/10.1109/TGRS.2021.3049921, 2021.

Geer, A. J.: Physical characteristics of frozen hydrometeors inferred with parameter estimation, Atmos. Meas. Tech., p. to be submitted, 2021.

Gong, J. and Wu, D. L.: Microphysical properties of frozen particles inferred from Global Precipitation Measurement (GPM) Microwave Imager (GMI) polarimetric measurements, Atmos. Chem. Phys., 17, 2741–2757, https://doi.org/10.5194/acp-17-2741-2017, https://www.

110  atmos-chem-phys.net/17/2741/2017/, 2017.

Khvorostyanov, V. I. and Curry, J. A.: Thermodynamics, kinetics, and microphysics of clouds, Cambridge University Press, https://doi.org/10.1017/CBO9781139060004, 2014.

Klett, J. D.: Orientation model for particles in turbulence, J. Atmos. Sci., 52, 2276–2285, https://doi.org/10.1175/1520-0469(1995)052<2276:OMFPIT>2.0.CO;2, https://journals.ametsoc.org/view/journals/atsc/52/12/1520-0469_1995_052_2276_omfpit_

115  2_0_co_2.xml, 1995.

Noel, V. and Sassen, K.: Study of planar ice crystal orientations in ice clouds from scanning polarization lidar observations, J. Appl. Meteorol., 44, 653–664, https://doi.org/10.1175/JAM2223.1, https://journals.ametsoc.org/view/journals/apme/44/5/jam2223.1.xml, 2005.

Prigent, C., Defer, E., Pardo, J. R., Pearl, C., Rossow, W. B., and Pinty, J.-P.: Relations of polarized scattering signatures observed by the TRMM Microwave Instrument with electrical processes in cloud systems, J. Geophys. Res., 32, https://doi.org/10.1029/2004GL022225,

120  https://agupubs.onlinelibrary.wiley.com/doi/abs/10.1029/2004GL022225, 2005.

Spencer, R. W., Goodman, H. M., and Hood, R. E.: Precipitation retrieval over land and ocean with the SSM/I: identification and characteristics of the scattering signal, J. Atmos. Ocean. Technol., 6, 254–273, https://doi.org/10.1175/1520-0426(1989)006<0254:PROLAO>2.0.CO;2, https://journals.ametsoc.org/view/journals/atot/6/2/1520-0426_1989_006_0254_prolao_2_0_co_2.xml, 1989.

---

## Author Response (AR2)

Dear Joseph Munchak,

The authors would like to give their thanks to the reviewers for their time and efforts in reviewing our submitted manuscript. We also thank them for their constructive comments and suggestions that certainly have improved the manuscript.

Please find attached our revised manuscript and response to the anonymous referee #3. This document contains, besides this page, the response to the comments of the anonymous referee #3 followed by the marked-up manuscript highlighting the revisions made to the manuscript. The revised manuscript is submitted as a separate file.

Most manuscript changes are done in direct response to the comments of the anonymous referee #3. These are discussed on the following pages. Other changes include some pure language corrections or other minor textual changes that improve the clarity of the manuscript.

Best regards, Vasileios Barlakas, Alan J. Geer, and Patrick Eriksson

**Response to Referee #2**

To begin with, we would like to thank the anonymous referee #3 for her/his time and efforts in reviewing our submitted manuscript and providing feedback. We would like to inform the anonymous referee #3 that we will revise the manuscript according to her/his constructive comments. Below we respond to the main questions/comments raised by the referee, and outline how we will revise the manuscript. To that end:

**• referee's comments are given in blue,**

- our responses are outlined in normal format, and
- any suggested textual changes are given in bold format.

**Responses to specific comments (GC) from referee #3 (GC3)**

10 (GC3.1) L82 – The "cloud overlap scheme" has been mentioned several times in the text, and is considered as a fast approximation of sub-grid variability. I understand that the scheme is introduced by Geer et al. (2009), but I think authors need to give a brief description of this method in the text.

Following the comment of the reviewer, we will introduce an extra equation and suggest the following revision:

**15**

The simulated  $T_{\rm B}$  is calculated as the weighted mean of the  $T_{\rm B}$  from two independent columns linked to clear-sky ( $T_{\rm B,clear}$ ) and cloudy ( $T_{\rm B,cloud}$ ) conditions:

$$T_{\mathbf{B}} = (1 - C_{\mathbf{w}}) \cdot T_{\mathbf{B}, \mathbf{clear}} + C_{\mathbf{w}} \cdot T_{\mathbf{B}, \mathbf{cloud}}.$$
(1)

Where Cw is the effective cloud fraction representing the vertical mean of cloud and precipitation fractions weighted
by the hydrometeor content (Geer et al., 2009a, b). This is considered to be a fast approximation of sub-grid variability and hence, the beam-filling effect (e.g., Barlakas and Eriksson, 2020).

(GC3.2) L87–88 – What is the difference between "convective snow" and "large scale snow"? Can large scale snow be considered as stratiform snow? The convective snow is assumed to be graupel, how the density of graupel is defined in the simulations?

The reviewer raises a very good point. We should clarify that large scale snow describes precipitating frozen hydrometeors in stratiform clouds, while convective snow refers to frozen hydrometeors within convective cores. The names "snow" and

25

"graupel" are just convenient labels for these representations within RTTOV-SCATT. As described in the paper in Table 1, the snow particle is represented by an ARTS sector snowflake and the convective snow particle is represented by a Liu 3-bullet

30 rosette. Hence, there is no direct concept of density (as there would be with a spherical soft Mie particle). Instead the layout of ice is defined by the 3D particle representation used in the DDA calculations.

To avoid any misconceptions, we propose to revise the manuscript as follows:

- At L87–88,
- First, instead of a four hydrometeor configuration (rain, snow, cloud liquid water, and cloud ice water), it separates convective snow (ice hydrometeors in deep convective cores; hereafter graupel) from large scale snow (precipitating ice hydrometeors in stratiform clouds; hereafter snow) leading to a five hydrometeor convention; a feature not available in v12.3. Further clarifications are given in Sect. 2.3.
  - At the end of Sect. 2.3, around L156,
- 40

Convective rain and snow are diagnostic variables, derived from a mass flux convection scheme that assumes the convective cores occupy only 5% of each grid box.

(**GC3.3**) L100-102 – "In case the retrieval fails, the TELSEM is employed." What do you mean by "retrieval fails"? Please specify it. A short explanation of Karbou et al. (2010) and the TELSEM is needed.

45

Following the reviewer's comments, we propose to revise the manuscript as follows:

The land surface emissivity is primarily retrieved from satellite observations in surface sensitive channels following the emissivity retrieval approach described in Karbou et al. (2010). This approach is based on the assumption that surface

- 50 emissivity varies only slowly with frequency. Accordingly, surface emissivities are first retrieved in window channels at lower frequencies and then these values are used as the surface emissivity for nearby sounding channels. This approach was extended to all-sky assimilation by Baordo and Geer (2016). However, the retrieval can be unreliable in strongly scattering scenes, if the cloud and precipitation is inconsistent between model and observations; it can generate a surface emissivity outside the physical bounds of 0 and 1, and even if within those bounds, the retrieval must be further
- 55 quality checked against values from an atlas. The atlas values are from the Tool to Estimate Land Surface Emissivities at Microwave (TELSEM) (Aires et al., 2011), which is a monthly average emissivity climatology constructed from 10 years observations from the Special Sensor Microwave/Imager (SSM/I). If the emissivity retrieval is out of physical bounds or far from the value in the atlas, the atlas value is used instead.

60 (GC3.4) L107-108 – The search of the "optimal" microphysical setup was based on SSMIS. Could you please specify what do

The "optimal" microphysical setup was found by optimising measures of fit between actual observations from SSMIS and the equivalent SSMIS observations simulated by the Integrated Forecast System (IFS) of ECMWF (see Eq. 1 of the manuscript).

- The choice of SSMIS was made on the basis that it is one of the few instruments covering most of the frequencies presently utilized in all-sky data assimilation (Geer et al., 2017, 2018) and it has been already employed in parameter estimation studies (Geer and Baordo, 2014; Geer, 2021). However, SSMIS with mostly H-channels at high frequencies (above  $\approx 150.0$  GHz) will generally see bigger scattering depressions than sensors with mostly V-channels at similar frequencies (as GMI). Consequently an iterative procedure was used: Geer (2021) first created an "intermediate" microphysical configuration without the use of the
- 70 polarization scheme. That configuration was used here to derive the best polarization ratio. Then, Geer (2021) included our polarization scheme in order to derive a new and self-consistent final configuration for RTTOV v13.0; this final microphysical configuration was extremely similar to the "intermediate" configuration in terms of bulk optical properties, demonstrating the effectiveness of this approach.
- 75 Following the referee's comments we propose to include the following clarifications:
  - At L107-108,

The search of the "optimal" microphysical setup was treated as a cost minimization problem between actual observations and simulated observations from the Special Sensor Microwave Imager/Sounder (SSMIS).

- At the end of Sect. 2.2,
- 80

85

Note here that the polarized scattering correction scheme that is introduced later in Sect. 2.5 was used by Geer (2021) in order to derive a final and fully consistent microphysical configuration for RTTOV v13.0.

(GC3.5) L118-119 – It claims that "The PSD introduced by Field et al. (2007) is a good physical representation for snow and further employed to represent graupel." Could you please provide references? Snow and graupel dominate different cloud regimes (stratiform vs convective), could you please explain why Field et al. (2007) is a good for both cases?

The reviewer is absolutely right; our positioning could be misleading. The particle size distribution (PSD) introduced by Field et al. (2007) was constructed on the basis of in situ observations of tropical anvils and stratiform clouds at mid-latitudes. This implies that it should not be the best physical choice for the rather large hydrometeors found within deep convective cores.

90 As far as large scale (stratiform) snow concerns, the Field et al. (2007) PSD is a well established PSD (Kulie et al., 2010; Fox et al., 2019; Fox, 2020).

In Sect 2.2, we described that the optimal microphysical setup identified by the multi-dimensional parameter search Geer (2021) resulted from the bulk scattering signature, i.e., the combination of the PSD and the hydrometeor type (with its explicit

 $\alpha$  and  $\beta$  coefficients of the mass-size relation). That said, although, the morphology of the selected hydrometeors and/or the

- 95 PSD may not be considered as the most physically correct representations, their combination gave the best fit between the actual observations and the simulated counterpart. Note here that Geer (2021) also tested the PSD by Marshall and Palmer (1948), which produces larger particles. However, it was found not to be the best option within the context of data assimilation, since it led to a degradation of the fit between observations and simulations.
- 100 Following the reviewer's comment, we propose the following revision:

The PSD introduced by Field et al. (2007) (tropical configuration) was retained as a **good representation for snow within** the context of DA (e.g., Geer and Baordo, 2014; Fox, 2020) and, was similarly found as a good option to represent graupel in the IFS.

**105**

**(GC3.6) L125 - Could you please give references to which observations are you referring to?**

The search of the "optimal" microphysical setup was based on SSMIS as outlined in L107-108. Following the referee's comment we will repeat this information in L125. To that end, we suggest the following revision:

[revised manuscript text omitted]

$$PD^{o} = T^{o}_{BV} - T^{o}_{BH},$$

$$PD^{b} = T^{b}_{BV} - T^{b}_{BH}.$$
(7)

- 310 One of the challenges encountered was to screen out any surface contamination and particularly, any strong polarization signal that originates from water surfaces (Meissner and Wentz, 2012); strongly polarized surfaces complicate the separation between cloudy and clear sky conditions. Appendix A describes a number of careful screening checks that were used to minimize the surface contribution. Note here that throughout Sect. 3, results are presented in terms of the screened data that are almost entirely free from surface contribution.
- Figure 2 illustrates the global PD as a function of TBV at 166.5 GHz in case of both observations and simulations. Polarized scattering from preferentially oriented ice hydrometeors leads to PDo up to 10–15 K (in red) centered around 220 K, with increasingly low TBV indicating increasingly cloud-affected scenes. In fact, the arch-like shape of the PDo ToBV (or ToBH) relation appears to be universal (Gong and Wu, 2017). The existing modelling framework characterized by a polarization ratio of 1.0 (control run; in green) completely fails to reproduce such polarization signal (see Fig. 2a). However, it provides valuable information regarding the surface contribution; if there was any simulated polarization signal due to ocean reflection, it would
- be visible. The remaining panels in Fig.2, i.e., b–f, depict the ability of  $\rho$  to provide realistic simulations of this behaviour, and a first glance indicates that a  $\rho$  value between 1.2 and 1.4 could do a reasonable job, since it is within the distributions of the observations.

**3.2 Quantifying the fit of model to observations**

325 In order to quantify the best fit between simulations and observations, the commonly used metrics are the mean, the standard deviation ( $\sigma$ ), and/or the root mean square error (rmse). Nonetheless, forecast modelling systems are still unable to predict cloud and precipitation at small scales, and thus, are characterized by mislocation errors that could lead to quite large  $\sigma$  and/or rmse (e.g., Geer and Baordo, 2014, and references therein). A more comprehensive assessment of the polarized scattering